# Barbarigenesis and the collapse of complex societies: Rome and after

**Doug Jones** *

Department of Anthropology, University of Utah, Salt Lake City, Utah, United States of America

* douglas.jones@anthro.utah.edu

**Data Availability Statement:** Annotated Mathematica files and pdfs for models of wealth-power mismatch in 2- and n-player games, and data on collapse of social complexity, Rome and after, are available online through the Open Science Foundation (OSF), DOI 10.17605/ODF.IO/HUFXV.

## Abstract

"Barbarism" is perhaps best understood as a recurring syndrome among peripheral societies in response to the threats and opportunities presented by more developed neighbors. This article develops a mathematical model of *barbarigenesis*—the formation of "barbarian" societies adjacent to more complex societies—and its consequences, and applies the model to the case of Europe in the first millennium CE. A starting point is a game (developed by Hirshleifer) in which two players allocate their resources either to producing wealth or to fighting over wealth. The paradoxical result is that a richer and potentially more powerful player may lose out to a poorer player, because the opportunity cost of fighting is greater for the former. In a more elaborate spatial model with many players, the outcome is a *wealth-power mismatch*: central regions have comparatively more wealth than power, peripheral regions have comparatively more power than wealth. In a model of historical dynamics, a wealth-power mismatch generates a long-lasting decline in social complexity, sweeping from more to less developed regions, until wealth and power come to be more closely aligned. This article reviews how well this model fits the historical record of late Antiquity and the early Middle Ages in Europe both quantitatively and qualitatively. The article also considers some of the history left out of the model, and why the model doesn't apply to the modern world.

## Introduction

Societies past and present differ in their level of social development [1–3]. More developed societies may or may not provide improved biological well-being, greater prosperity, or more liberty and equality for most of their members [4, 5]. But they do consistently command more resources, and excel in their "abilities to master their physical and intellectual environments and get things done in the world" [1]. Social development is correlated with social scale, social complexity, and intensity of economic production. As a rule, scale, complexity, and intensity tend to increase together over time and to diffuse in space from more to less developed regions. These long-term, large-scale trends result from multiple causes, including population pressure, capital accumulation, intellectual and practical innovation and diffusion, growth in the extent of the market and the division of labor, and competition between social groups [1, 3, 6–9].

**Funding:** The author received no specific funding for this work.

**Competing interests:** The author has declared that no competing interests exist.

That's the general rule. But here we are concerned with understanding some exceptions to the rule. More specifically we are concerned with how a process in time—the collapse of complex societies—may result from a process in space—*barbarigenesis*, the formation of "barbarian" societies. Let's consider time and space in turn, and then in conjunction.

## Complexity through time

Social complexity doesn't always increase over time. Societies often stagnate. Societies sometimes decline, with the trend toward complexity going into reverse. Some reversals proceed to the point of enduring social collapse, a "rapid significant loss of an established level of social complexity" [10]. Things fall apart:

> Extending across different domains of human activity, from the economic to the intellectual sphere, [collapse] typically results in diminished social stratification, social differentiation and division of labor, the abatement of flows of information and goods, and a declining investment in civilizational features such as monumental architecture, art, literature, and literacy. These developments accompany and interact with political disintegration . . . In severe instances, population as a whole contracts, settlements shrink or are abandoned, and economic practices regress to less sophisticated levels [5].

The collapse of complex societies has attracted both popular and scholarly fascination [10–12]. The large literature on the subject is concerned not just with what happened but *why*. There are many possible *why*s; probably no single cause operates in every instance of collapse, and most collapses have multiple causes. Nonetheless, there are some recurring patterns. We are concerned here with one such pattern, which arises from the interactions between complex societies and their less complex neighbors.

## Complexity in space

Social complexity tends to diffuse in space, as complexity in one area begets complexity in neighboring areas. The common result is a spatial gradient from an advanced core to a laggard periphery. But social evolution in the periphery does not, in general, simply recapitulate evolution at the core. Social complexity is multi-dimensional, an imperfectly correlated bundle of traits—economic, military, political, and cultural. These traits may diffuse at different rates, as the opportunities and dangers of living with wealthier, more powerful neighbors deflect peripheral societies away from a single line of across-the-board increasing complexity and along a special path.

In this article, we develop a line of argument about one special path, its causes and consequences. In the next section, the argument is presented as a formal model. Here we offer a preliminary sketch.

Societies differ not only in their level of social development and the resources they command, but in how they allocate resources in the face of tradeoffs. Every society faces a tradeoff between plowshares and swords—between producing economic goods ("wealth"), and coercively appropriating the goods that others produce and protecting their own goods from appropriation ("power"). This tradeoff is recognized in current economic theory as a distinction between productive activity, which increases total wealth, and unproductive rent-seeking, which aims to redistribute wealth; resources invested in rent-seeking result in a net social loss [13]. Rent-seeking is ubiquitous in stratified state societies: ruling elites and predatory states extort *protection rent* from producers, much of which is invested in military establishments that defend against rival rent-seekers [14, 15].

We are particularly interested here in wealth-power tradeoffs that play out in space, where the allocations that societies make to production and/or predation/protection depend both on their own resources and on the resources and allocations of their neighbors. In the next section we introduce a simple model of conflict over the division of wealth, developed by Hirshleifer, in which the weaker of two players enjoys a paradoxical advantage. We then elaborate a multi-player spatial model, in which, as a result of actors' interdependent choices, wealth and power diffuse unevenly from core to periphery. The result is a *wealth-power mismatch*. The core comes to have a greater share of wealth relative to military power, while their neighbors on the periphery come to have a greater share of military power relative to wealth.

The model is meant to capture a recurring historical phenomenon: the formation of "barbarian" societies adjacent to more complex societies. A word about terminology: the labels "barbarian" and "barbarism" (and their cognates in other languages) have been used with a variety of meanings, sometimes referring pejoratively to uncouth outsiders, sometimes referring to a proposed stage of social evolution intermediate between "savagery" and "civilization." Many scholars today avoid these terms altogether, or use them only inside scare quotes. But others employ them more freely, and Scott has recently made the case for a respectable particular conception of barbarism. In his telling, barbarism is a special path, a recurring syndrome among folk interacting with a more complex society.

> "Barbarians" are . . . not a culture or a lack thereof. Nor are they a "stage" of historical or evolutionary progress . . . "[B]arbarian" is best understood as a position vis-à-vis a state or empire. Barbarians are people adjacent to a state but not in it [16].

This understanding of barbarism is particularly suited for our purposes. Following Scott, we do not label a society as barbarian simply because it falls within a certain level of social or political development. Instead, we reserve the term for peripheral societies that develop a wealth-power mismatch—a relative excess of power over wealth—in the course of interacting with more complex societies, as spelled out in the next section. We call the process that produces these societies *barbarigenesis*. Not all peripheral societies will count as barbarian in this sense; the Conclusion to this article discusses why, as a result of changes in the nature of military power, barbarigenesis has been less characteristic of the modern world system.

## Complexity in space and time

Following our initial static model of barbarigenesis, we proceed to add a time dimension. In a model of historical dynamics, a wealth-power mismatch leads to rent-seeking and collateral damage, which produce a long decline in social complexity, sweeping from more to less developed regions, until wealth and power come to be more closely aligned.

We apply the model to one particular stretch of space and time. The geographic area is Europe, the time span broadly 1 to 1200 CE, or more narrowly 200–1000. This covers the decline and fall of the Roman Empire in Europe, first in the west, then in the east, the continuing decline of social complexity through the early Middle Ages, and the beginnings of a Europe-wide recovery. (The different fate of the Roman empire outside Europe, and its neighbors and successors there, falls outside our discussion.)

A few general remarks about model building are in order. Models generally face a tradeoff between realism, precision, and generality [17]; this model comes down on the side of generality, with a corresponding sacrifice in precision and realism. We have more to say below about what is left out when we compare the model's predictions with the course of history. But there

are advantages to generality as well. Consider this bird's eye view of how Europe changed over 1000 years. In 1 CE.

> the European landscape was marked by extraordinary contrasts. The circle of the Mediterranean . . . hosted a politically sophisticated, economically advanced and culturally developed civilization. This world had philosophy, banking, professional armies, literature, stunning architecture and rubbish collection. Otherwise, apart from some bits west of the Rhine and south of the Danube . . . the rest of Europe was home to subsistence level farmers, organized in small scale political units. . . . The further East you went, the simpler it all became [18].

> But "move forward a thousand years, and the world had turned" [18] After a long drawn out collapse in social complexity and population, Europe had begun to recover. Social complexity was now more evenly distributed in space. A system far out of equilibrium, with radical geographic disparities, had given way to a more homogeneous one. This levelling process was not a simple wave of advance of social complexity. The most developed regions had declined, while the least developed had advanced.

> Of course a huge amount of history happened over this interval—the rise and fall of a number of empires and kingdoms, episodes of partial recovery and further decline, multiple devastating epidemics, and the birth and expansion of the world's two major proselytizing religions. Our model doesn't account for all these developments, and also leaves out many details of geography and ecology. The last part of the Discussion has more to say about these factors. There we argue that models and analyses of these complications largely complement rather than contradict the present effort. Specifically, insofar as the various political and ecological crises of late Antiquity and the early Middle Ages had exceptionally severe and enduring consequences, it was owing in large part to the presence of sizable, powerful barbarian groups on the periphery, a rolling reservoir generated by contact with the core.

## Methods: Models

> What is the paradox of power? It is the seemingly puzzling observation that poorer or weaker contenders often gain from conflict, at the expense of richer or stronger opponents. . . . [T]he battle is not always to the strong [because] in a wide range of circumstances it pays the smaller or weaker contender *to fight harder* [19].

> Wars are caused by undefended wealth.

> Attributed to Ernest Hemingway.

Economic theory is most typically concerned with mutually beneficial exchange. But the methods of economics—standard assumptions about maximizing payoffs—can also be used to analyze more conflict-ridden transactions, like lawsuits, strikes, and wars. Here we start with a model of conflict between two players.

### The paradox of power

A simple model of conflict can have some interesting outcomes. Consider the following game (in the game theory sense), devised by Hirshleifer [19]. Each of two players, 1 and 2, has some level of resources, $R_1$ or $R_2$, at his disposal. (We assume throughout that these are resources over and above subsistence.) Each player can allocate his resources to production ($E_1$ or $E_2$) or

to fighting ($F_1$ or $F_2$), so that

$$R_1 = E_1 + F_1 \quad \text{for player 1,} \tag{1}$$

and

$$R_2 = E_2 + F_2 \quad \text{for player 2,} \tag{2}$$

with all $R$, $E$, and $F$ non-negative.

The sum of the production of the two players, $E_1 + E_2$, is divided between them based on a *conflict success function*. The function we use here depends on the ratio of the fighting efforts of each, so that players' incomes, $I_1$ and $I_2$, are given by

$$I_1 = F_1{}^m/(F_1{}^m + F_2{}^m) \cdot (E_1 + E_2) \tag{3}$$

$$I_2 = F_2{}^m/(F_1{}^m + F_2{}^m) \cdot (E_1 + E_2) \tag{4}$$

Clearly $I_1 + I_2 = E_1 + E_2$.

The exponent $m$ in the equations is a parameter that determines the decisiveness of conflict, the degree to which a greater fighting input, $F_1/F_2$, translates into a greater income, $I_1/I_2$. With $m = 1$, a player who fights twice as hard ($F_1/F_2 = 2$) gains twice the share of income ($I_1/I_2 = 2$). With $m = 2$, a player who fights twice as hard gains $2^2$ or four times the share of income ($I_1/I_2 = 4$). In the rest of this article, we assume that $m = 1$. This is the simplest case, and there are also reasons for thinking that it is realistic for ancient warfare. For modern warfare a higher value of $m$ may be more appropriate [20]; we discuss some implications for barbarigenesis in the Conclusion. (In Hirschleifer's model, the two players' productive efforts may be complementary, so their total production amounts to more than the sum of their productive efforts. Here we assume zero complementarity.)

If each player tries to maximize his income, given the actions of the other player, the equilibrium allocations to production and fighting will depend on the resources of each. Where the two players have equal resources, the equilibrium allocation has each player allocating ½ his resources to production and ½ to fighting.

In this case, when the ratio of initial resources, $R_1/R_2$, is 1, the ratio of final incomes $I_1/I_2$ is also 1. But when initial resources are unequal, the initial disparity in resources results in a smaller disparity in incomes, or no disparity at all. Consider for example the case where player 1 has twice the resources of player 2, 10 units versus 5, say. (The ratio of resources is what matters; the absolute units are arbitrary.) In this case, the equilibrium allocations are $F_1 = F_2 = 3.75$ and incomes are $I_1 = I_2 = 3.75$ The weaker player in other words, is devoting just as much effort to fighting as the stronger payer, and far less to production. With equal resources devoted to fighting, the two players also have equal incomes.

This outcome demonstrates in stark form the *paradox of power*. Greater resources mean greater *potential* fighting ability, but don't translate into greater *realized* fighting ability because greater resources also mean a higher opportunity cost for choosing fighting over production—swords over plowshares. For the weaker player, the opportunity cost of fighting is less. This player stands to gain from putting a greater fraction of his resources into fighting than in the equal-resources case, and to specialize in extracting rents from the stronger.

When the disparity in resources is great enough, the equilibrium strategy for the weaker player is to abandon production altogether and specialize entirely in fighting (a corner solution). With $R_1 = 10$ for the stronger player, this point is reached when $R_2 = 3.33$ for the weaker player. At this point, $F_1 = F_2 = 3.33$ and $I_1 = I_2 = 3.33$.

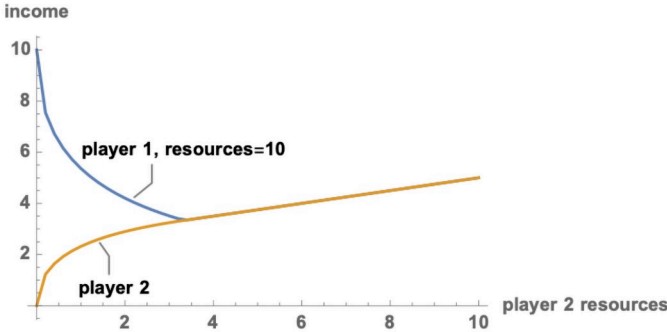

**Fig 1. The paradox of power.** Incomes (resources-minus-fighting-efforts) in a two person game for player 1, resources = 10 (arbitrary units), and player 2, varying resources.

The outcome of this game might be seen as an exercise in comparative advantage. But it is a perverse kind of comparative advantage which does not operate to the mutual benefit of the two parties. Fig 1 shows income accruing to each player when the first player has resources $R_1 = 10$ and the second player has resources $R_2$ varying from 0 to 10. The figure might be taken as an initial demonstration of the potential for barbarigenesis: a society of intermediate complexity on the periphery will specialize in rent-seeking from the core, and the core will do worse than it would in facing either a stronger or a weaker opponent.

## Power and wealth in space: Barbarigenesis

To get to our model of barbarigenesis, we begin by considering what the game looks like with more than two players.

There are several ways to generalize from the two player model above to an $n$ player model. In one $n$ player game, all the wealth produced by the players is thrown into a common pool [19]. The contents of the pool are shared out among the players, with each getting a share dependent on his fighting effort. In this version of the game, there is less and less incentive to produce anything as the number of players increases and each player gets a smaller fraction of total production. As $n$ approaches infinity, production approaches zero and fighting effort consumes all resources.

This model of wealth allocation as an anarchic free-for-all is not useful for our purposes. Instead of having all players contend together for shares of a common pool, we consider a game where each of $n$ players faces multiple two-way contests. Suppose player $i$ has resources $R_i$ at his disposal. He allocates $F_{ij}$ to fighting player $j$ in a two-way contest. His total fighting effort, summed over all $j$, is then $\Sigma_j F_{ij}$, and he allocates the remainder of his resources to production, $E_i$, so $E_i + \Sigma_j F_{ij} = R_i$. As before, all $R$, $E$, and $F$ are non-negative.

We assume that in the two-way contest between $i$ and $j$, $i$ gets a share of $j$'s production, $E_j$, that is a function of (1) the effort $i$ puts into fighting $j$, $F_{ij}$, (2) the effort everybody puts into fighting j, $F_{jk}$ for all $k \neq j$, and (3) the effort j puts into fighting everybody, $F_{kj}$ for all $k \neq j$. As before, the decisiveness of the conflict depends on the parameter $m$, so defining $I_{ij}$ as the income that $i$ takes from $j$, we have

$$I_{ij} = F_{ij}{}^m / (\Sigma_k F_{jk}{}^m + \Sigma_k F_{kj}{}^m) \cdot E_j \tag{5}$$

In addition, the share of player $i$'s own production, $E_i$, that he hangs onto, $I_{ii}$, is a function of (1) the total effort he puts into fighting everyone, $F_{ij}$ for all $j \neq i$, and (2) the total effort everyone

puts into fighting him, $F_{ji}$ for all j$\neq$i. So

$$I_{ii} = \Sigma_j F_{ij}{}^m / (\Sigma_j F_{ij}{}^m + \Sigma_j F_{ji}{}^m) \cdot E_i \tag{6}$$

and $i$'s total income, $I_i$, is the sum of what he takes from others and what he hangs onto,

$$I_i = \Sigma_j I_{ij} + I_{ii} \tag{7}$$

As in the two person case, total income is equal to total production, $\Sigma_i I_i = \Sigma_i E_i$.

In one version of this game, all $n$ players have equal resources, say 10 units. Above we consider the case where $n = 2$, and the optimal E and F are 5 and 5 for each player. With more than two players, the optimal allocation changes somewhat. More resources go to fighting because when $i$ puts more resources into fighting $j$, he not only does better in his two-way contest with $j$, but also gets a larger share of the total plundered from $j$. However, in contrast to the free-for-all game with a common pool of resources, the optimal level of production never goes to zero. Instead, as $n$ goes to infinity, the optimal E and F for each player go to 4 and 6 asymptotically.

For our purposes, modeling barbarigenesis, we are more interested in cases with uneven resources. As an illustrative example, Table 1 shows equilibrium outcomes when $n = 3$ and the resources of players 1, 2, and 3 are 10, 4, and 3 respectively. We observe that players 2 and 3 put no effort into fighting one another because player 1 is a more profitable target. This cease-fire between the weaker players holds when the sum of their resources is less than 8 or 9. Player 1 puts less effort into fighting players 2 and 3 than they put into fighting him, because he divides his fighting effort, and players 2 and 3 each get less income than player 1, because they divide the spoils.

For the more general case with $n = 3$, Fig 2a shows the income for a strong player ($R = 10$), when the resources of the two other players range from 0 to 10. Fig 2b shows the income for one of the weaker players. As in the two player case, the strong player fares badly when facing players with intermediate resources. His worst outcome is even worse than in the two player case.

To develop our model of barbarigenesis in space, we will want to calculate optimal allocations for a number of players—more than three—of varying resources at different locations. However for $n > 3$ and players with unequal resources, it becomes difficult to compute players' optimal allocations. We cope with this by sticking with three person games, and averaging over a number of such games. Imagine, then, a set of players, $n_x$ in number, strung out evenly along a line in space. For a given player, we calculate the income from each of the three person games that he could play with two other players; these games number $(n_x - 1)(n_x - 2)$. That player's total income is then an average of these incomes. We assume that contests are more

**Table 1. Outcomes in a three person game, an illustrative example.**

|  | Player 1 | Player 2 | Player 3 |
|---|---|---|---|
| Resources, R | 10 | 4 | 3 |
| Fighting effort, F |  |  |  |
| Player 1 | 0 | 2.85 | 2.21 |
| Player 2 | 2.51 | 0 | 0 |
| Player 3 | 1.71 | 0 | 0 |
| Production, E | 5.78 | 1.15 | .79 |
| Income, I | 3.51 | 2.39 | 1.82 |

The 3x3 Fighting effort matrix shows how much effort each Column player puts into fighting each Row player.

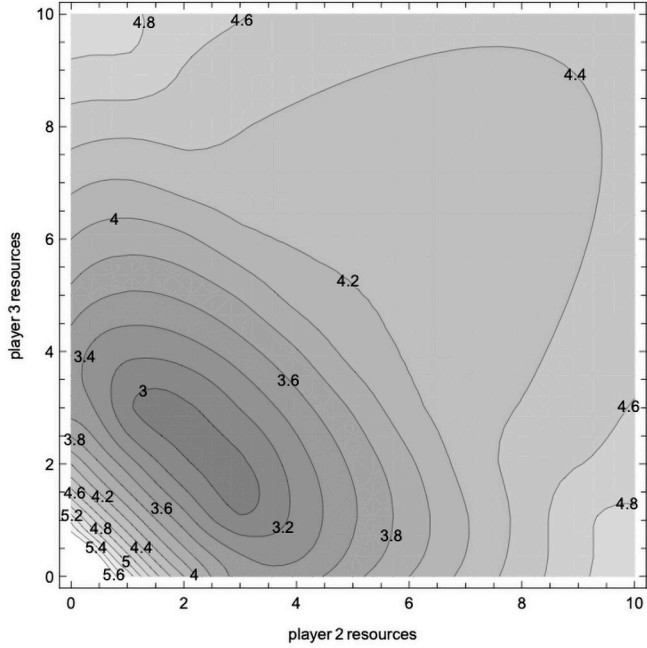

**2 a**

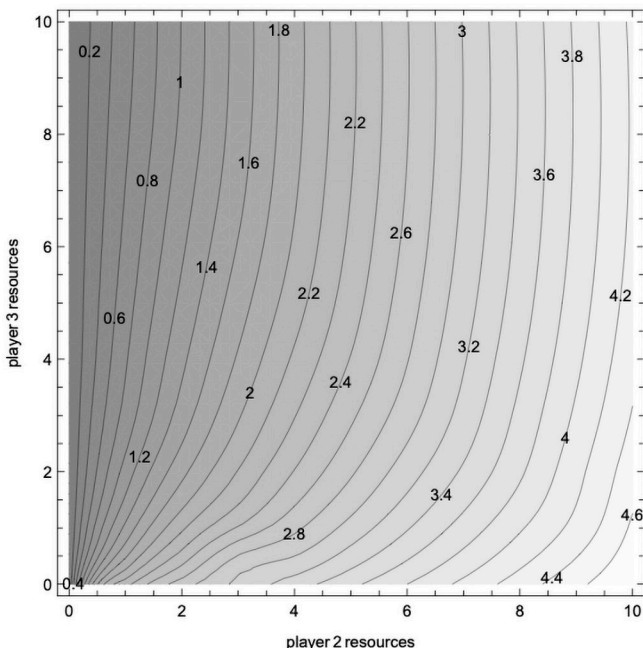

**2 b**

**Fig 2. The paradox of power with 3 players.** a. Income (contour lines) for player 1, resources = 10, facing players 2 and 3, varying resources. Player 1's income is 10 when resources are 0 and 0 for players 2 and 3. His income is lowest, 2.95, when resources are 2.33 and 2.33 for players 2 and 3, who are putting everything into fighting. His income is 4.44 when resources = 10 for all players. Bottom edge corresponds to upper curve in Fig 1. b. Income (contour lines) for player 2, varying resources, facing player 1, resources = 10, and player 3, varying resources. Bottom edge corresponds to lower curve in Fig 1.

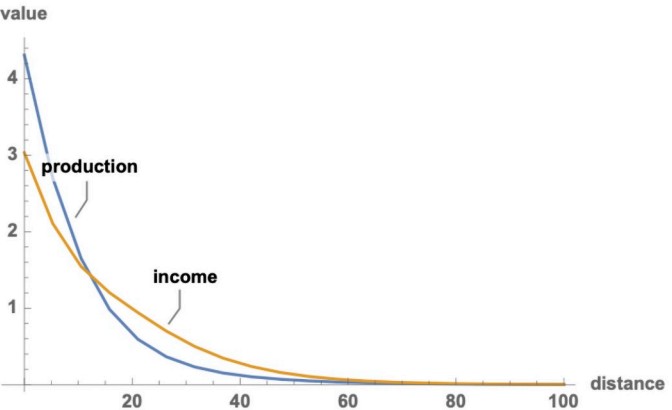

**Fig 3. Wealth-power mismatch.** Average outcomes for all possible combinations of 3 person games among 20 players along a line in space (arbitrary distance units; games among nearby players weighted more heavily). Resources fall off exponentially with from core to periphery. Central players produce more than they get; more distant players get more than they produce.

frequent with nearby than distant players, so the average is weighted according to how close the other players are. This means that the income, $W_i$, of player $i$ is given by

$$W_i = \text{Mean}[W_{ijk} \cdot (1 - D_{ijk})^2] \tag{8}$$

where $W_{ijk}$ is the income accruing to $i$ in a three person game among $i$, $j$, and $k$, assuming each of the three plays to maximize his income, and the function Mean is the mean taken over all $j$ and $k$, $i{\neq}j{\neq}k$, weighted by $(1-D_{ijk})^2$ where $D_{ijk}$ is the mean of the Euclidean distances of $i$ to $j$ and $i$ to $k$. Furthermore, we let $R_i$ be the resources available to $i$, and $P_i$ be the total fighting effort expended by $i$, averaging as above over all the three person games that $i$ plays. This means that $i$'s total production is $E_i = R_i$-$P_i$, which may be greater or less than his income. Finally, we convert the discrete variables $R_i$, $W_i$, $P_i$, and $E_i$ over i = 1,. . ., $n_x$ (with $n_x = 20$) by interpolation into continuous functions $R$, $W$, $P$, and $E$ over $x$, where $x$ is spatial distance in arbitrary units.

Now suppose resources fall off from an advanced core to a less developed periphery, and players allocate their resources optimally between production and fighting as described above. With resources falling off exponentially in space, the result is as shown in Fig 3. Wealth and power both decline as we travel from the core to periphery, but not at the same rate. The result is a *wealth-power mismatch*, with the core allocating a relatively greater share of its resources to wealth production, and the periphery allocating relatively more to fighting. This is our initial model of barbarigenesis.

## Wealth and power over time: From barbarigenesis to decline and fall

We can go from the static model above to a dynamic model by making $R$, $W$, and $P$ functions of both space, $x$, and time, $t$. To begin with, suppose the rate of growth in resources, $\partial R /\partial t$, at a given point in space and time is given by

$$\partial R/\partial t = r \cdot R \cdot (k - R) + a \cdot \partial^2 R/\partial x^2 \tag{9}$$

This is the standard wave of advance model, which has been used to model everything from the spread of an advantageous mutation in space [21], to the range expansion of an invasive species [22], to the spread of a new mode of subsistence through demic expansion [23, 24]. On

the right, the first term is an expression for *logistic growth*, where resources in one place grow at a pace set by the intrinsic rate of increase, *r*, but limited by the carrying capacity, *k*. The second term is an expression for *diffusion*, with resources diffusing from regions of greater concentration to neighboring regions of less concentration, with the parameter *a* giving the rate of diffusion. If the growth of resources were governed by this equation, the resulting dynamics would be a wave of advance moving from an initial region of greater concentration to regions of less concentration, with the wave asymptotically approaching a constant roughly sigmoidal shape and a velocity of $2(r \cdot a)^{\frac{1}{2}}$.

We have seen above how, for a given distribution of resources in space, we can calculate the optimal allocation of resources between production and fighting at each point. Given Eq 9, allocations to production and fighting would look like time-shifted versions of the wave of advance in resources. Fighting effort would increase in advance of increasing resources, as societies with intermediate-to-low resource levels turned to extracting resources from their wealthier neighbors. Productive effort would lag behind: eventually those societies with high-to-intermediate resource levels would react to the increasing opportunity costs of fighting by shifting to relatively more production.

Eq 9 implies that, although a wealth-power mismatch shows up in the *allocation* of resources between production and fighting, the mismatch doesn't affect the *availability* of resources. The result is a model of barbarigenesis, but not collapse. To get a model of collapse, we add a further assumption: there is not only rent-seeking, an *un*productive diversion of resources into contests, but also collateral damage, a *counter*productive loss of resources.

Let's add to Eq 9 a term for collateral damage resulting from wealth-power mismatch:

$$\partial R / \partial t = r \cdot R \cdot (k - R) + a \cdot \partial^2 R / \partial x^2 - c \cdot \text{Ramp}[M] \cdot \text{Lag}[t] \qquad (10)$$

where *c* is a parameter giving the vulnerability to collateral damage, Ramp is a function defined by Ramp[*z*] = *z* for $z \geq 0$ and Ramp[*z*] = 0 for $z < 0$, and Lag[*t*] is a sigmoidal function going from ~0 at *t* = 0, to .5 at *t* = 300, and to ~1 at *t* = 600, with *t* in units of years CE.

At the center of the collateral damage term is an expression for wealth-power mismatch, *M*. The wealth-power mismatch, $M_{ijk}$, in a game among players *i*, *j*, and *k* is defined by

$$M_{ijk} = E_{ijk} - W_{ijk} - .1R_{ijk} \qquad (11)$$

As above (Eq 8), we define $M_i$ as the mean of $M_{ijk}$ over all *j* and *k*, $i \neq j \neq k$, weighted by (1-$D_{ijk})^2$. If the amount *i* produces is less than or roughly equal to his income, this number is negative. If the amount he produces is greater than his income by a significant amount, this number is positive, and he suffers collateral damage. (The .1$R_{ijk}$ term means that collateral damage doesn't kick in if players are fairly similar.) In other words, not only is net income shifted away from those whose wealth exceeds their power, but some of their productive resources are lost in the process.

The collateral damage term also incorporates a time lag. We have to assume that barbarigenesis involves some kind of time lag in order to allow for the initial accumulation of resources. Here we assume a sigmoidal function that starts near zero in the year 1, and reaches near 1 in the year 600. This implies that wealth-power mismatch is present from the beginning, but becomes more destructive over time.

## The barbarian frontier in space and time

A verbal summary of the argument so far: the course of social evolution often results in the formation of a core area rich in resources and a poorer periphery. But the periphery is not just the core with fewer resources. The threats and opportunities posed by rich neighbors may result in

barbarigenesis, with the periphery devoting a greater share of its resources to fighting rather than production. The economic logic behind the paradox of power can be set out in a game with just two players, while the geography of barbarigenesis can be framed as a collection of games laid out in space (either discrete or continuous space). In this case, we see a wealth-power mismatch: central regions have comparatively more wealth than power, more peripheral regions have comparatively more power than wealth, and the most distant regions have little of either.

Core and periphery may evolve over time. In the simplest case, there is a wave of advance in resources from center to periphery. Proceeding in advance of this wave is a wave of investment in fighting, lagging behind is a wave of investment in production. In a more complex case, wealth-power mismatch results not merely in net transfers of wealth, but in collateral damage and destruction of resources—in the collapse of social complexity.

These assumptions can be embodied in a simple model. The main variable is resources, which vary as a continuous function of space and time. Resources are optimally allocated at each moment between production and fighting. Resources tend to increase over time, to diffuse in space, and to suffer collateral damage where the gap between wealth and power is especially great. We assume an initial exponential distribution of resources, and a time lag in the onset of collateral damage. There are just a few adjustable parameters: the intrinsic rate of increase in resources, $r$, the rate of spatial diffusion of resources, $a$, and the vulnerability to collateral damage, $c$.

That's it: one dimension of space; no explicit boundaries between polities; no internal organization to societies. Obviously this is a drastic simplification of reality. We will see below that it nonetheless seems to capture some central processes operating on the *longue durée*.

## Methods: Data

In the next section we use population density as a proxy for resources, and find model parameters that give a best fit for population estimates for Europe from 1 to 1200 CE.

We use population estimates from McEvedy and Jones [25]. These numbers are useful for our purposes as they represent a serious attempt to produce population estimates at two century intervals for each of nineteen European countries over the whole period we consider. Other efforts cover more limited areas or time periods, or treat Europe or broad areas of Europe as a whole at longer intervals. However, the limitations of these figures need to be noted. They were compiled more than forty years ago, reflected in the fact that several of the countries (Yugoslavia, Czechoslovakia, and "Russia" i.e. the Soviet Union in Europe) no longer exist. And because the estimates are made at two century intervals, they necessarily miss population fluctuations on shorter time scales, e.g. from before to after the Antonine plague (165 CE).

In the Results and Discussion sections we deal with some of these issues as they come up. (The notes for the Mathematica notebooks supply more information.) In the rest of this section, we compare McEvedy and Jones' numbers to some other broad estimates.

Table 2 presents some population estimates for the Roman empire in Europe. Frier [26] gives higher numbers than McEvedy and Jones, with more vigorous population growth. Harper's [27] numbers are higher still. However the ratios of different regional populations are more similar.

Table 3 compares another set of estimates, for a longer period, over Europe as a whole. Livi Bacci [28] assumes a higher population and more vigorous growth in the early centuries, but the broad pattern of growth, decline and recovery is similar.

**Table 2. Population estimates for the Roman empire, by European region (thousands).**

|  | McEvedy& Jones | | Frier | | Harper |
| --- | --- | --- | --- | --- | --- |
|  | 1 CE | 200 CE | 14 CE | 164 CE | 164 CE |
| Italy | 7,000 | 7,000 | 8,100 | 8,700 | 14,000 |
| Iberia | 5,000 | 5,500 | 5,000 | 7,500 | 9,000 |
| Gaul/Germany | 5,750 | 7,500 | 5,800 | 9,000 | 12,000 |
| Danubian lands | 3,050 | 3,550 | 2,700 | 4,000 | 6,000 |
| Greece | 2,000 | 2,000 | 2,800 | 3,000 | 3,000 |

McEvedy and Jones [25], Frier [26], Harper [27].

**Table 3. Population estimates for Europe (millions).**

| Year | McEvedy & Jones | Livi Bacci |
| --- | --- | --- |
| 1 | 31 | 41 |
| 200 | 36 | 55 |
| 400 | 31 |  |
| 600 | 26 | 31 |
| 800 | 29 |  |
| 1000 | 36 | 41 |
| 1100 | 44 |  |
| 1200 | 58 | 64 |

McEvedy and Jones [25], Livi Bacci [28].

Table 4 gives estimates of per capita income for the Roman world from a recent review [29] synthesizing earlier work. The figures (which are very uncertain) generally show greater income per capita in areas and periods of greater population density.

Table 5 gives four indices of social development for 1–1200 CE, for the developed core of Western Eurasia—the portion of West Eurasia tied together by the densest political, economic, social, and political interactions, from Morris [1]. Most of the indices show the same pattern of decline and recovery as in the other tables, although the index of social organization (the size of the largest city), is more volatile, and partly reflects political contingencies independent of overall economic development [1].

In summary, any discussion of population and resources during this period must acknowledge great uncertainties about the numbers [35]. For the Roman period, we have some census figures, but there are consequential disagreements between "low count," "middle count," and

**Table 4. Per capita income for the Roman world, by European region.**

| Year CE | 14 | 150 | 300 | 400 | 500 | 520 | 600 | 700 |
| --- | --- | --- | --- | --- | --- | --- | --- | --- |
| Peninsular Italy | 2.14 |  | 1.35 | 1.31 | 1.18 | 1.15 | 1.11 | 1.13 |
| Gaul / Iberia | 1.20 |  | 1.35 | 1.31 | 1.18 | 1.15 | 1.11 | 1.13 |
| Roman Britain | 1.10 |  | 1.22 | 1.20 | 1.04 | 1.04 | 1.04 | 1.07 |
| Aegean world | 1.30 |  | 1.31 | 1.31 | 1.35 | 1.37 | 1.31 | 1.11 |
| Total empire | 1.42 | 1.81 | 1.33 | 1.32 | 1.28 | 1.29 | 1.24 | 1.15 |

Estimated mean per capita income as a multiple of subsistence income. Milanovic [29], based on Allen [30], Goldsmith [31], Maddison [32], Scheidel and Friesen [33], Ward-Perkins [34].

**Table 5. Four indices of social development in the west.**

| Year CE | Energy capture | Organization | War-making capacity | Information technology |
|---|---|---|---|---|
| 1 | 31,000 | 1,000 R | .12 | 4.29 |
| 100 | 31,000 | 1,000 R | .12 | 4.29 |
| 200 | 30,000 | 1,000 R | .11 | 4.29 |
| 300 | 29,000 | 800 R | .10 | 2.98 |
| 400 | 28,500 | 800 R | .09 | 2.98 |
| 500 | 28,000 | 450 Cn | .07 | 2.98 |
| 600 | 26,000 | 150 Cn | .04 | 1.65 |
| 700 | 25,000 | 125 Cn | .04 | 1.65 |
| 800 | 25,000 | 175 B | .04 | 1.65 |
| 900 | 25,000 | 175 Cd | .05 | 1.65 |
| 1000 | 26,000 | 200 Cd | .06 | 2.30 |
| 1100 | 26,000 | 250 Cn | .07 | 2.30 |
| 1200 | 26,500 | 250 B/Ca/Cn | .08 | 3.60 |

From Morris [1].

Energy capture is energy consumption, food and non-food, in daily kilocalories per capita. Social organization is population, in 1000s, of largest city.

R Rome, Cn Constantinople, B Baghdad, Cd Cordoba, Ca Cairo.

War making is an index of military power, relative to 250 for the USA in 2000.

Information technology is equal to the sum of Information Technology Points (.5 for full literacy, .25 for medium, .15 for basic) times percent of population.

"high count" scholars about whom the census was counting [36–38]. For the later period we have even less to go on. At the same time, for our purposes these limitations are not absolutely critical. There is less disagreement about population ratios between different times and places than about absolute numbers. And our use of population density as a proxy for wealth, while it may understate the economic differences, probably does so in a consistent fashion. In any case, fitting our model to a different set of estimates would probably yield somewhat different parameters, but qualitatively similar results.

## Results

We want to find the combination of parameter values, $r$, $a$, and $c$, that gives the best fit to the population estimates.

To begin with, for initial conditions in the year 1 we use the values shown in Fig 3. To get the vertical scale we assume that the most densely settled region has a population of .6 of carrying capacity, $k$. This is the ratio of Italy's population at the height of Roman wealth to its population in the high Middle Ages and early modern period, according to McEvedy and Jones [39].

Given the initial conditions and carrying capacity, we want to assess the goodness of fit for a given $r$, $a$, and $c$. For country $i$ at time $t_j$, our measure of fit is

$$\left( M\left[i, t_j\right] - u \cdot W\left[x_i, t_j\right] - v \right)^2 \qquad (12)$$

where $M[i, t_j]$ is population density according to McEvedy and Jones, and $W[x_i, t_j]$ is wealth in space and time according to the model. The measure of overall goodness of fit is the mean of this quantity over 19 countries and 8 time points. The variables $u$, $v$, and the nineteen $x_i$ are chosen to minimize this measure. This means that $x_i$ is country $i$'s *ascribed distance*, a measure of $i$'s central versus peripheral position.

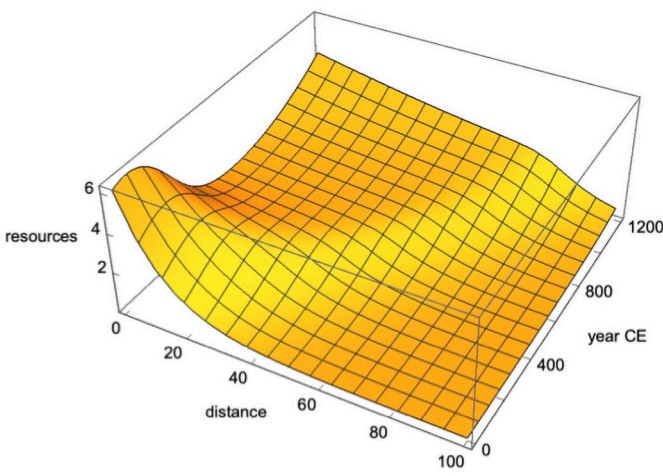

**4 a**

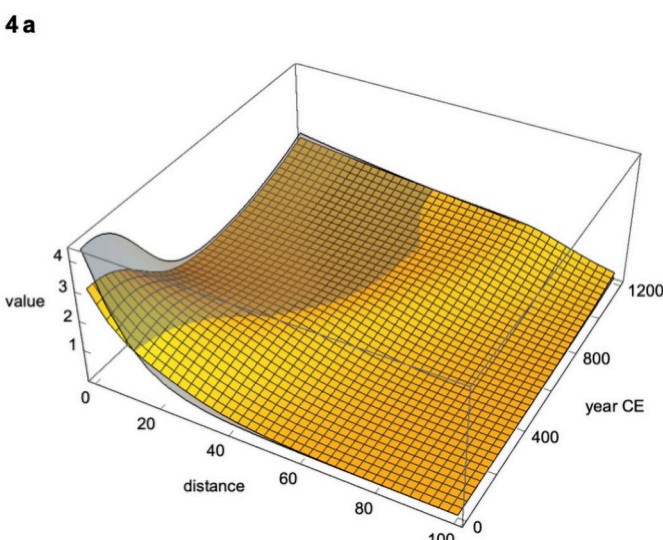

**4 b**

**Fig 4. From barbarigenesis to collapse to recovery.** a. Resources. Starting in the year 0 with resources falling off exponentially with distance, resources tend to (a) increase logistically over time, (b) diffuse in space, and (c) decrease over time where there is a positive wealth-power mismatch (after a time lag in the early centuries). Parameter values are chosen to give a least-squares best fit to population estimates. (In this and succeeding figures we start the first millennium with the year 0 rather than the year 1, which is more computationally convenient.). b. Production and income, given resources as in 4a. The gray surface is production, the tan surface is income. Initial values (year 0) equal production and income in Fig 3. Wealth-power mismatch spreads but ebbs over centuries.

When we vary the parameters to find a least squares best fit, we get $r = .009$, $a < .001$ (not distinguishable from 0), and $c = 8$. The mean square deviation from the model is 3.20, while the variance (the mean square deviation from the mean) is 28.5. The resulting curves for resources, production, and fighting are shown in Fig 4. After a period of stability or growth in the early centuries, resources decline in areas close to the core where wealth production exceeds income. Resources begin to recover from around the mid millennium, although these regions continue to display an excess of wealth over power. The opposite, an excess of power over wealth, is manifest just beyond the core, and spreads from there. By the end of the

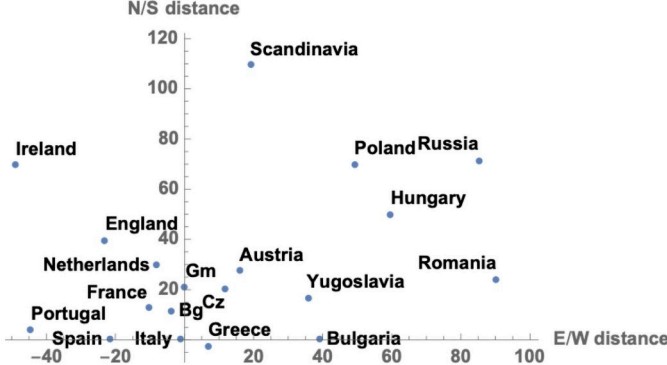

**Fig 5. Ascribed distance and geography.** The *direction* of each country relative to Italy is the actual geographic angular direction from Rome. The *distance* of each country from Italy is the distance ascribed by the model, in arbitrary units. Bg = Belgium, Gm = Germany, Cz = Czechoslovakia.

millennium, resources are widely diffused, and the collateral damage attendant on the wealth-power mismatch is less of a brake on growth.

Fig 5 shows how each country's one-dimensional ascribed distance, as assigned by the model, relates to the two-dimensional geography of Europe. Italy is at the center of the figure, and other countries are arrayed around it. The *direction* to each country outside Italy is the actual direction from Rome to the center of that country, but the *distance* of each country from Italy is the ascribed distance. Ascribed distances correlate with actual distances (correlation.60). The deviations from actual distances seem to make sense: waterways—the Mediterranean and Rhine particularly—pull some countries closer, and mountains—the Alps and Balkans particularly—push others away.

Fig 6 shows how each country's ascribed distance relates to the date, if any, at which it was incorporated into the Roman empire. The figure shows something about the economics of empire building: more developed countries were more attractive targets for early incorporation; less developed were likely to be incorporated later or not at all.

Fig 7 shows population estimates and model predictions over time for each country. A few comments here (with more to follow in the Discussion section):

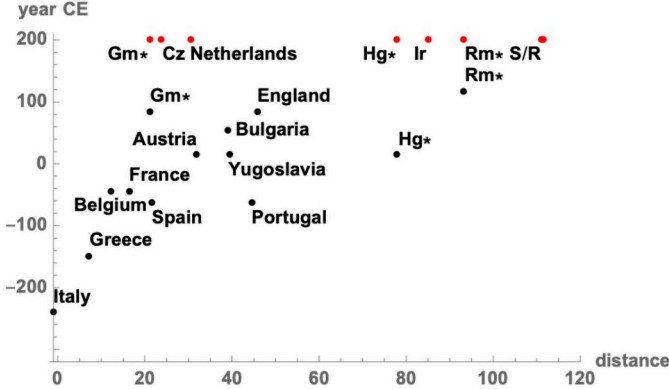

**Fig 6. Ascribed distance and the expansion of Rome.** The date for each country is the date at which most of the modern country was incorporated into the Roman empire. Countries never incorporated are assigned a nominal date of 200 CE. Countries incorporated at most partially appear twice, with *. Gm = Germany, Cz = Czechoslovakia, Hg = Hungary, Ir = Ireland, Rm = Romania, S/R = Scandinavia / Russia.

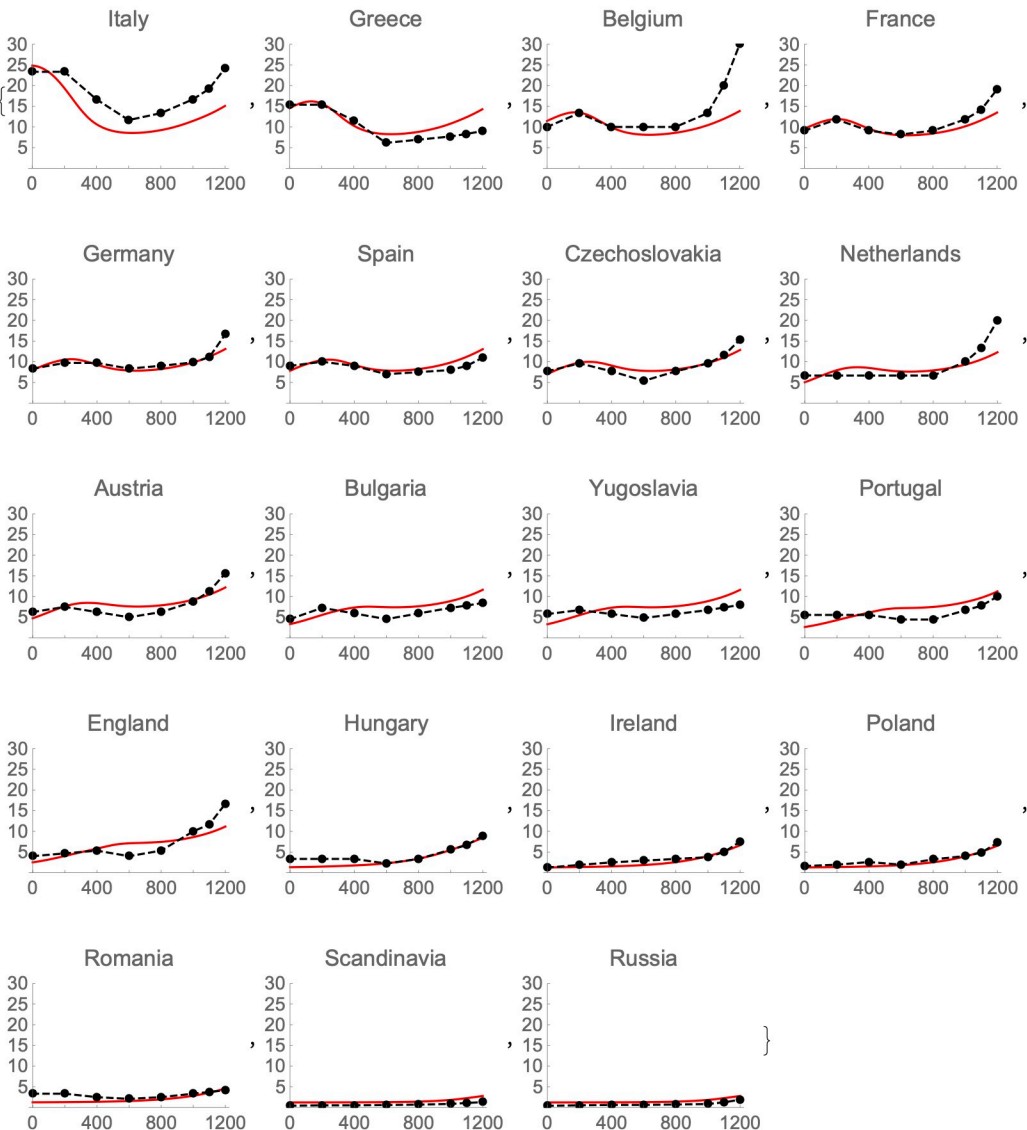

**Fig 7. Barbarigenesis and the fates of nations.** Population densities for 19 countries, estimated (points and dashes) and predicted by the model (solid lines). The x-axis is years CE, y-axis is population per square kilometer.

Italy fares better than predicted by the model after 1 CE, and several other countries, notably Belgium and the Netherlands, do a lot better than predicted in the two centuries after 1000. Such discrepancies are not surprising. Italy has a greater fraction of arable land than other countries in southern Europe. And, as the second millennium gets rolling, barbarigenesis and its aftermath become less important in determining outcomes, and other factors, like access to potential trade routes, become more important.

Greece does worse than predicted in later centuries. Greece is a special case for several reasons. First, the figures from McEvedy and Jones are particularly in need of revision here [40]. It now looks like Greece fared better than they reckon up to the mid sixth century CE, but then experienced a sharp collapse in population and wealth in the later sixth century, and an even weaker post-collapse recovery. Endemic disease, especially malaria, and resource exhaustion may be factors in later centuries. Greece had already experienced a loss of population by the

Roman era, after a peak, probably in the fourth century BCE. In other words, the history of Greece—developed, literate, and urbanized before Italy, declining with the rise of Rome, reviving with the rise of Byzantium, stagnating later—reflects environmental and political factors beyond those considered here.

## Discussion

In the Models section, we develop a simple model of barbarigenesis and its consequences. A mismatch in wealth and power between core and periphery generates a spreading, enduring collapse in social complexity, followed by a broad recovery. In the Data and Results sections, we show that the model provides a good fit to population estimates for Europe over the first millennium—a reasonable proxy for social complexity.

While the quantitative fit is encouraging, another test of the model is qualitative: how well does it fit the historical record? Here we address this question in several stages. First, we argue that the model's picture of *what* happened, especially of decline and fall and the role of barbarians, has broad scholarly support. Then we consider some features of the model including opportunity costs, rent-seeking, and collateral damage. These operate in part, we argue, through a variety of intermediate processes, including the establishment, internal evolution, and collapse of states and state boundaries, and invasions and migrations. We spell this out, and then consider how these intermediate processes, as well as some geographic and ecological variation left out of the model, also complicate things.

### Decline and fall and the barbarians

There is broad, if not unanimous [41], scholarly agreement that there was a collapse in social complexity, a "decline and fall," in Europe in the first millennium CE.

> The overall economic trend of the Roman world from c. 200 to 700 was downward. This is not to say that decline prevailed everywhere, all the time. . . . But . . . the overarching [downward] pattern is now clear, even if the details are sometimes sporadic and even contradictory [39].

The archeological record shows this clearly [34]. A recent review declares:

> One thing that archaeology makes very clear . . . is the dramatic economic simplification of most of the West . . . Building became far less ambitious, artisanal production became less professionalized, exchange became more localized. The fiscal system, the judicial system, the density of Roman administrative activity in general, all began to simplify as well [42].

There is significant regional variation in the extent and timing of decline (more on this below). To what extent the collapse of social complexity entailed a decline in health and physical well-being for the survivors is uncertain. A smaller population may have experienced improved nutrition in some cases, and a more rural population may have borne less of a burden of endemic crowd diseases [43]. Economic inequality declined along with the decline and fall, although mostly owing to losses at the top rather than to gains at the bottom [5, 29]. But the decline in population, in production above subsistence level, and in social complexity over a wide area are firmly established.

There is also broad, if not unanimous, scholarly support for the idea that conflict between core and periphery played an important role in the collapse.

I would not argue that there were any inherent instabilities in the Roman state that would explain its collapse in the West in the fifth century. This is by now not a controversial position: the majority of scholars would argue that the period around 400 was one of institutional stability and also economic prosperity. . . . Such a position would thus put considerable stress on the 'barbarian' invasions as the major catalyst that resulted in the end of the western empire [44].

The Empire was always hampered by its economic, political and administrative limitations, but there is not the slightest evidence that it would have ceased to exist in the fifth century without the new centrifugal forces generated by the arrival of large, armed immigrant groups [18].

Even in the west, the fall of the empire *caused* the decline, and not vice versa. There were structural weakness and human blunders, as ever, but it is no easy calculation to make these add up to an event as momentous as the disappearance of central imperial power in the west [27].

During the same centuries that the most developed areas of Europe saw a dramatic collapse in social complexity, government of these areas passed from the Roman empire to multiple successors, ranging from extensive kingdoms to rougher small-scale polities, ruled by the descendants of barbarian invaders.

In our model, the causal arrow goes in one direction: pressures from a militarized periphery—barbarian plunder, invasion, conquest, and settlement—cause a rolling collapse in the core. The reality was not always so simple. Decline in the core was not *only* a result of external forces. Causation could run in the opposite direction, with weakness in the core—the result of internal troubles, political and ecological—inviting invasion (more on this below). Nevertheless, the evidence is that barbarians were important agents of decline and fall. They were not simply arriving in the aftermath of collapse to pick up the pieces.

It is the finer details of decline that provide some of the best evidence for the importance of the barbarian factor. In the model, collapse and recovery happen smoothly, and this provides a good fit to the data if we look at long term trends. But the long term averages mask short term ups and downs. A brief summary:

The historical record shows limited external challenges, civic peace, and relative prosperity from the establishment of the Principate (27 BCE) to the late second century. At the same time, the sheer number of barbarians increased considerably through the early centuries CE. Population growth in barbarian Europe was both extensive and intensive. Between the first and third centuries, west Germanic groups transitioned from shifting cultivation to intensive cultivation with heavy plows and manuring of fields, and settled in more stable villages. East Germanic groups would follow after a lag of several centuries [18, 45, 46]. After 400 CE the Korchak and Penkovka archaeological cultures, probably ancestral to later Slavs, abandoned scratch plows for more productive heavy plows [47].

Barbarian pressure on the empire seems to have intensified from the late second century. From the late second century through the late third the Empire passed through a multi-dimensional political, economic and epidemiological crisis. In mid-third century, a flood of barbarian invasions and attacks reached deep into the Empire. Several areas—a slice of Germany, Dacia—were abandoned to barbarians. Subsequent developments within the Roman empire look like responses to increased barbarian threat, although other factors may have been involved. Beginning in the third century, the empire went through a spate of wall-building [48]. "During the great invasions of the second half of the third century and later, many town

walls were built in both the eastern and western provinces, often in great haste, and from demolition debris and gravestones" [49].

The late third century saw the establishment of a new political order, the Dominate, and relative political stability continued until late in the fourth century [50]. The army expanded. "Between the early third and mid-fourth centuries the 300,000 strong Roman army increased in size by at least one third, and quite possibly by substantially more" [51]. An expanded military required higher levels of taxation, and the imperial finances went through a major overhaul under Diocletian. Tax rates increased substantially, and tax collection was largely removed from the control of local optimates to a salaried bureaucracy. The disposition of the military changed in important respects: the *limitanei*, troops guarding the frontier were supplemented with *comitatenses*, field armies stationed inside the borders of the empire, with the latter getting the best troops. On one interpretation, this looks like a change in strategy in response to an increased military threat, a change which exposed the civilian population of the interior to increased threat [52], (but see [53]).

Finally, in the fifth century the empire in the West disappeared entirely in the wake of barbarian invasions.

The archeological record in the West seems to track these changes. Trends in Italy, Britain, Belgica, Gaul, and Spain are broadly similar. Rural settlement density increases under the Principate, declines somewhat in the third century in some areas, partially recovers in the fourth, and collapses in the fifth [54]; see also [55, 56]. A detailed look at settlement densities in different regions suggests the importance of the barbarian factor. Belgica and north Gaul experienced barbarian invasion in the third century and show a concomitant loss of population. Settlement density recovered only slightly in the fourth century, and the frontier continued to be "scarred and thus uninviting" [48]. Population in this area collapsed in the fifth century with further invasions and the fall of the Empire. Britain and south Gaul, less affected by barbarian invasion in the third and fourth century, passed through this period largely unscathed. Britain experienced a sharp decline in the fifth century with the collapse of Roman rule. In Italy and south Gaul, the survival of Roman institutions under barbarian rule was stronger, and the fifth century collapse milder. North Africa, spared the initial invasions, flourished for a time, but experienced a sharp decline with the arrival of the Vandals in the fifth century [51].

In summary, the crisis of the third century resulted from a number of factors, with barbarian invasions playing an important but not exclusive role. The invasions hit some provinces harder than others, and this shows up in the archeological record. The recovered empire—the Dominate—was greatly changed. Some of the changes—wall-building, military expansion, higher taxes—were arguably a direct response to increased barbarian pressure, although an increased threat from the Sassanian state, Rome's eastern rival, was also important. Other changes—an apparent decline of the middle class and increase in inequality—may have been partly an indirect consequence of the reorganization of the empire in the face of external threats, although internal social developments and responses to other crises, notably major epidemics, also played a part. However there is no reason to think that internal causes alone would have brought the empire down. The final collapse of the empire in the west, and the accompanying collapse in population and economy, was directly tied to barbarian invasions.

In the east, in the Balkans and Greece, events played out differently. We say more about the separate path of the east below, but for now we note that although the timing was different, the core of southeastern Europe did eventually experience a dramatic collapse in social complexity, with barbarian invasions playing a central role.

## Opportunity costs

So far we have been considering *what* happened. But our model also implies something about *how* it happened, how a relatively small number of barbarians had an oversized impact. Specifically, in the model, the greater resources available to folk in the core under imperial rule are counterbalanced by their greater opportunity cost of fighting. This seems to be consistent with several scholarly analyses.

> The Roman Empire simply became too expensive for its inhabitants, who were no longer willing to pay in blood and money for its military power [57].

Measuring resources by population and economic production, the core had a great advantage over its neighboring periphery. "There can be little doubt that the empire possessed considerably greater reserves of manpower than the barbarians" [58]. In the fifth century, when the Roman Empire fell in the west, "historians generally propose up to 100,000 for major ruling groups like the Ostrogoths or the Vandals, and around 20,000–25,000 for the adult males who made up their armies, in provinces whose indigenous populations numbered in the millions" [42]; also [57]. There were similar disproportions in the sixth century, when much of Italy fell to the Lombards, and in the ninth and tenth centuries, when Vikings raided and settled in northern Europe. Even where the barbarian fraction was arguably greater—Franks in north Gaul, Angles, Saxons, and Jutes in Britain, Slavs in the Balkans—they were still in the minority.

In other words, if all parties had realized their full military potential and put all their economic surplus into fighting, the imperial core would easily have come out ahead. But recall the paradox of power: "the battle is not always to the strong [because] in a wide range of circumstances it pays the smaller or weaker contender *to fight harder*" [19]. Applied to the present case, the paradox implies that what determined outcomes in the contest between rich core and poor periphery was not just the absolute resources of each, but the opportunity costs of fighting and preparing to fight.

Economic concerns were central both in imperial expansion and contraction, even if the parties involved were not keeping careful accounts, or undertaking explicit profit maximization. "The Roman emperors had at least a crude sense of the 'marginal costs of imperialism'" [27]. When the Roman empire was expanding, the dates at which different regions were incorporated into the empire corresponded with their economic potential (Fig 6).

During periods of decline as well, considerations of costs and benefits were crucial. The Roman military suffered some major defeats, notably at Adrianople (378 CE) where the emperor Valens and two thirds of his army perished. But the more fundamental cause of the fall is that the cost of defense came to exceed what people were willing to pay. Already under the Dominate the empire offered less bang for more bucks: citizens found themselves paying higher taxes and (probably) getting less military protection. Contrary to earlier views [59], high taxes and bureaucracy do not seem to have crippled the economy, but they did undermine support for the empire [58].

In the late third and fourth centuries, the empire confronted multiple invasions, from Visigoths, Ostrogoths, Vandals, Suevi, Alans, Burgundians, and Franks. The invaders were sometimes bought off with grants of territory and a status as *foederati*; more often they forcibly seized what they wanted. In any case, when territory ravaged or occupied by barbarians was lost as a source of revenue, the army could no longer be paid. In less than a century the Roman empire in the West unraveled completely. After two more centuries, the empire in southeastern Europe unraveled. No decisive battle ended the empire; it became unaffordable [51].

Even after the collapse of imperial rule the old Roman elite did not just disappear: some were killed, some fled, but others remained and adapted to the new regimes. Some even flourished, although on terms dictated by their new barbarian overlords [60].

Different scholars offer differing assessments of Roman-barbarian relations in the transition. On one account, it was mostly about the art of the deal: "What we call the Fall of the Roman empire was an imaginative experiment that got a little out of hand" [61]. A more somber judgment comes from Ward-Perkins [34]: "The Germanic invaders of the Western empire seized or extorted through the threat of force the vast majority of the territories in which they settled, without any formal agreement on how to share resources with their new Roman subjects." From our perspective, these quotations point to flip sides of the paradox of power. On one side, the paradox implies that "non-conflictual or cooperative strategies tend to be relatively more rewarding for the better-endowed side" [19], and the Roman empire, Roman elites after the fall, and Roman successor states, all showed themselves willing sometimes to bargain and collaborate with barbarian intruders. On the other side, violence and the threat of violence from those with less to lose played a determining role in the transition. The game between Romans and barbarians was not zero-sum, but it was a long way from purely cooperative.

## Varieties of rent-seeking: States and migrations

Our model predicts that where there is a wealth-power mismatch between core and periphery, there will be rent seeking. The exact mechanisms are not specified by the model, but they included, at different periods, shifts in power and wealth within the Roman empire, raiding and plunder, the consolidation of barbarian confederations and kingdoms, and barbarian invasion, migration, and mass settlement. We review varieties of rent-seeking below.

Our model implies that even in the early stages, when collateral damage is slight, we should see evidence of wealth-power mismatch. In the context of the early centuries CE, this means we expect to find a mismatch *within* the Empire, with more developed regions increasingly specializing in producing wealth and less developed regions increasingly cultivating a military specialization.

Even before the establishment of the Principate, this dynamic was at work, as Rome extended her rule over the Mediterranean. During this period, the eastern Mediterranean, ruled by Hellenistic monarchs, was more economically developed than the Roman West. This both made the area an inviting target for conquest, and contributed to military weakness: Eastern militaries were largely mercenary, and expensive. Rome at this point depended on a cheaper army of citizen soldiers [62, 63].

With the establishment of the Principate, the military basis of the empire shifted to a professional soldiery committed especially to defending the frontier, often relatively removed from the civilian population. About 2/3 of state revenues, some 2–3% of gross domestic product, went to the military [27]. About half the army consisted of citizen legionaries, about half of auxiliary forces, mostly non-citizens. The regular army increasingly drew its men from the provinces, outside Italy and the more developed east.

[M]ost legionaries across the empire were of Italian origin until the reign of Claudius (AD 41–54). Through the reigns of Claudius and Nero, about half were Italian and half of provincial origin. By Trajan's reign (AD 98–117), legionnaires from the provinces outnumbered Italians by four or five to one [64].

The auxiliary forces too came to be largely drawn from the provinces: "It is by the blood of the provinces that the provinces are won" (Tacitus in [65]).

Barbarians may have come to make up an increasing fraction of the military [66, 67], (but see [68]). "The spatial, social, and ethnic peripheralization of military service—a feature common to many maturing empires—not only raised the profile of frontier forces but also drew in manpower from beyond" [63]. They became increasingly numerous in the higher ranks. "By the latter half of the fourth century increasing numbers of senior officers appear with 'barbarian,' frequently Germanic names" [69]. In the last days of the empire in the west, supreme military command increasingly passed to generals of Germanic origin, like Arbogast, a Frank, and Stilicho, a Vandal.

Political changes accompanied the demilitarization of the imperial core and the militarization of the periphery. Emperors from Trajan and Hadrian on found themselves spending increasing amounts of time close to the frontier, and the effective capitol shifted from Rome to Milan (286 CE) and then to Ravenna (402 CE). The old Senatorial elite of Italy, the *clarissimi*, continued to be extremely wealthy, but were edged out politically by a new senatorial elite. The crisis of the third century and subsequent recovery partly reflected these changes. In the third century, military units on the frontiers vied with Rome, putting up a bewildering succession of barracks emperors. Eventually a more settled situation developed as one frontier region, Illyria, came to monopolize the imperial succession.

These changes within the empire can be seen as the working out of the principle of comparative advantage, with more and less developed regions coming to specialize in production and fighting respectively. This was not the conventional, peaceable version of comparative advantage. These developments, resulting from wealth-power mismatch, were about rent-seeking: capturing wealth and forestalling its capture.

In subsequent centuries, with barbarian resources increasing outside the empire, barbarian rent seeking, trading on barbarian military prowess, is increasingly evident. This took a variety of forms. Military service, raiding and plunder, and the extortion of tribute, carried out by barbarian groups of various sizes at the expense of wealthier targets, are amply attested before and after the fall of Rome. There were also changes in social organization. Barbarian polities along the frontier probably increased their size and degree of organization, and grew more formidable. "There are clear signs that some barbarian units, especially just beyond the frontier were increasing in power and stability during the fourth century" [58]. Larger groupings appearing in the early centuries CE include the Franks ("Free/wild people"), Marcomanni ("Border men"), and Alamanni ("All men") [57]. These changes arguably resulted from the pressures and opportunities associated with proximity to a wealthy core. The changes were driven by trade and combat—offensive and defensive—with the Roman empire itself, and jostling among barbarians for access to imperial resources. They amounted, in short, to a phase of barbarigenesis. (This outline is widely but not universally accepted, see [18, 57, 58, 70, 71], but see also [69] and [72]. For a similar story of barbarian agglomeration and civilized response in the Viking age, with a dynamic model, see [73].)

Most dramatically and consequentially, barbarians could secure a share in the wealth of their neighbors by moving to where the wealth was. The first millennium has traditionally been seen as the Migration Period, the age of the *Völkerwanderung*. Below a few remarks on a large and disputed topic:

First, migrations during this period were mostly toward regions with denser population and greater wealth (with some exceptions, like the Norse settlement of Iceland). Some migrations proceeded from outer periphery to inner periphery. The Goths expanded from the Baltic area (Wielbark archeological culture) to the north shore of the Black Sea (Cernjakov culture) and took up plundering Roman territory on the farther shores. Huns, Avars, and Magyars moved from the Eurasian steppe to the grasslands of the Great Hungarian Plain and took up plundering and extorting tribute from the empire and its territory. Other groups moving from

outer to inner periphery at some point include Burgundians, Lombards, and Bulgars. Some migrations proceeded from the periphery to imperial or former imperial territory. Germanic peoples ended up ruling over most of the western empire, southern Slavs took over most of the eastern empire in Europe. Migrations were often interconnected. The early Gothic migration pressured west Germanic groups, the Huns pressured the Goths, and the Avars pressured the southern Slavs. Western Slavs moved into territory vacated by Germanic migrations. Both push and pull might be involved in the initial migration in a series. Avars, for example, were pushed to the western edge of the steppe by Turks. But the pull toward greater wealth stands out as the dominant theme in this period.

Second, the migrations entailed substantial costs. Most of the migrants were not habitual nomads. Moving to a new location, sometimes over very large distances, sometimes more than once, entailed a major reorganization of customary routines. Even for pastoral nomads, large scale moves into new territory were not an everyday occurrence. Migration could also entail challenging political transformations, including submission to new forms of authority.

Third, the migrations arguably involved the movements of large groups of men and women, not just elites or bands of soldiers. At least this is the traditional view [66], consistent with the writings of classical authors like Marcellinus Ammianus and Jordanes. However, this is an area of controversy; Halsall [58], for example, is a skeptic regarding large-scale migrations, while Heather [18] provides a nuanced defense of something closer to the traditional view.

In the future, new sources of evidence, especially studies of genetic variation, will advance this debate. For now, some preliminary results are available. The movement of Goths, including women, from the shores of the Baltic to the Black Sea is supported by genetic evidence [74], consistent with Jordanes, and contra Kulikowski [75] who argued for cultural transformation without major migration. The Anglo-Saxon invasions (unlike the later Norman invasion) had a substantial impact on the genetics of England [76, 77], contra the argument that Anglo-Saxonization involved only limited migration [78]. In sixth century Italy, ancient DNA from high status graves shows the central European affinities expected of Lombard invaders, while low status burials have local roots [79].

Thus the evidence to date suggests that at least some of the migrations of the *Völkerwanderung* were a real demographic phenomenon—less than population replacement, but more than culture shift. It looks like large groups of men and women from the barbarian periphery of Europe were paying the costs and enjoying the benefits of moving to, or close to, more central societies, and living off them.

## Collateral damage

Collateral damage from wealth-power mismatch *within* the empire was limited and episodic, with a partial recovery following the establishment of a new equilibrium under the Dominate. In a later period, as military advantage shifted further to the barbarians outside the empire, the damage would be increasingly severe and enduring. For the barbarian invaders of the Roman empire, the goal was to acquire Roman wealth, not to destroy it. Nevertheless, without anybody intending it, the first millennium saw a lasting collapse in social complexity and a decline in wealth, resulting to a large extent from the interactions between Europe's core and its periphery. As in our model, this happened because there was not only rent-seeking, an *un*productive diversion of resources into contests, but also collateral damage, a *counter*productive loss of resources.

Collateral damage, like rent-seeking, took a variety of forms. It was partly a matter of direct destruction of property and loss of life. Beyond this, the Mediterranean-centered trade

network collapsed, and the advantages of a Smithian economy, with an extensive division of labor were lost [39]. Perhaps most important, institutional breakdown and the insecurity of life and property must have discouraged individuals and groups from investing in the future.

The extent of collateral damage varied, depending on the character of political institutions. In some times and places, barbarians acted as *stationary bandits* [80]. A stationary bandit, in contrast to a roving bandit, has an incentive to preserve the long-term productivity of his targets. The itinerant armies of the *Völkerwanderung*–the Visigoths shifting around the Balkans, Italy, and Spain, the Suevi and Vandals moving through Gaul and Iberia—approximated roving bandits. By the end of the fifth century, however, these groups had settled down; most of the former Roman empire in the West was divided among a handful of successor states ruled by Germanic elites. Consistent with Olson's analysis, the new rulers were not purely predatory; they tried to maintain the traditions of Roman rule, and to enlist the collaboration of surviving Roman elites [81]. (See also [82] on the Vikings.)

The situation was complicated, however. The stationary bandits of the post-Roman world were not unitary actors [81]. The new rulers depended on the support of the barbarian rank-and-file, the military mainstay of the new kingdoms. These followers, the descendants of fractious unlettered warrior-farmers, were often unfitted and disinclined to play the role of obedient Roman-style subjects [83]. "[T]he Germanic tribes which broke apart the Western empire were not themselves capable of substituting a new or coherent political universe for it. The difference in 'water-levels' between the two civilizations was still too great" [84]. As a result, early barbarian kingdoms were hybrid regimes, with one legal system for Romans, another for Germans, with the latter enjoying a privileged position. The latter were also rewarded with a share of wealth at the expense of the former, either grants of land (the usual scholarly supposition [85, 86]) or a share of taxes [61]. In either case, central revenues were greatly reduced [87]. "Beginning in the fifth century, there was a steady trend away from supporting armies by public taxation and towards supporting them by rents derived from private landowning" [42].

In some cases, the balance between leaders and rank-and-file among the newcomers was weighted heavily toward the latter; enduring royal government was weak or nonexistent. This was particularly true where larger numbers of settlers moved shorter distances, as in Anglo-Saxon Britain, Frankish northern Gaul, and the Slavic Balkans. In these instances, the decay of Roman institutions and the decline in social complexity was particularly marked.

The collapse of social complexity in the first millennium was both cause and consequence of a decline in state capacity, collateral damage from the shift to a low-maintenance political regime that provided limited public order at a low price. "The new Germanic lords could not offer the same extensive administration to the landowners, but they did something else: they provided cheaper protection" [62]. Some early medieval kingdoms look impressive on a map, but "by the year 1000, [outside the Byzantine empire] it would have been difficult to find anything like a state anywhere on the continent in Europe" [88]. Early medieval polities are better described as realms than states [89]. Lasting recovery would wait until the rolling wave of barbarigenesis had subsided.

## Political complexities: States, cycles, and borders

The discussion up to this point has related political changes to the operation of large-scale forces over the long run. Shifts in wealth and power within the Roman empire, its dissolution first in the west and then in the European east, the formation of barbarian states, and their relative weakness when it came to maintaining law and order, all resulted, we argue, from the spatial dynamics of wealth-power mismatch, of wealth production and appropriation. But states

were not entirely at the mercy of larger forces; they could also be actors in their own right. Some complications resulted that fall outside our model.

In the model, decline and recovery happen smoothly. But internal factors, apart from the external stimuli we have considered so far, also contributed to the relative strength or weakness of states. In some cases these seem to have operated cyclically: the Roman empire in the west from the first to fifth century ran through a progression—stability, near collapse, partial recovery, collapse (see above)–that amounts to two up and down alternations, each lasting a few centuries. This cycle is superimposed on the long downward movement predicted by our model [90]. (The preceding rise and fall of Republican Rome is another up and down alternation.)

Boundaries between states, unrepresented in our model, also made a difference. This may be showing up in Fig 7, where there is a middle set of countries, from Bulgaria to England, for which the model is qualitatively somewhat "off." Take England. The model predicts modest growth to the mid-first millennium, stagnation as barbarigenesis among the country's less-developed neighbors takes a toll, and then recovery. The real story is more dramatic, probably even more dramatic, according to later research, than McEvedy and Jones' figures imply. As part of the Roman Empire from 43 CE, Britain experienced substantial prosperity and security. The withdrawal of Roman legions after 409 CE, followed by invasions from the Continent, resulted in a major—apparently catastrophic—decline in the density of settlement and the level of material culture [91]. "In no other part of the empire was this economic simplification so abrupt and total" [42]. In some respects, the level of material culture in Britain was lower after the legions left than before they arrived! It almost looks like the model is telling a story about an alternative history in which the Romans never occupied Britain. This counterfactual southern Britain, remaining outside the Roman Empire, avoids the wild swing from prosperity to utter collapse of the real-world England. In other words, being inside and then outside the Roman *limes* made a difference in ways not shown in our model. Similar observations apply to Bulgaria and Yugoslavia, where the later end of Roman rule was particularly devastating, and where, as in England, barbarian invasion led to language replacement. (Also related: in Germany there are differences right up to the present between areas that fell inside and outside the *limes* [92]).

These complications do not overturn the account given here, but they suggest that our model might usefully be supplemented by models of internally-driven secular cycles [90] and imperiogenesis [9]. These models do not capture the dynamics of barbarigenesis and its consequences that we explore here; they are complementary to the present effort.

## Geographic complexities: Rivers and mountains, steppe, and a sheltered zone

Other potential complications left out of our model involve geographic variation. In the Results section we run the model assuming uniform geography, and then look at deviations from the model as a secondary phenomenon. The deviations largely make sense, reflecting the impacts of waterways and mountainous terrain, and the availability of arable land (with Greece as something of a special case).

For much of the rest of Eurasia, the uniform geography assumption wouldn't work even as a rough first approximation. In the region from the Middle East through Central and South Asia to China, the division between steppe and desert on the one hand and areas of rain-fed and irrigation agriculture on the other hand was a decisive fact. Throughout this expanse, where pastoral nomads had agrarian states for neighbors, barbarigenesis took place, with the emergence of wealth-power mismatches, and the formation of states driven by external threats

and opportunities [93–95]. Because the border between periphery and core was the product of a climatically-dictated resource gradient between steppe and sown, the historical dynamics were different here from post-Roman Europe; this is a topic for another occasion.

Finally, where the geography of barbarians and decline and fall is concerned, the fate of the empire in the east seems to be the exception that proves the rule. In the Roman west—as in our model of barbarigenesis—the wealth of the core was up for grabs. In the east, by contrast, most of the core resource base was more secure. The European provinces of the eastern empire experienced a slew of barbarian invasions, by Visigoths (late fourth century), Huns (mid fifth century), and Ostrogoths (late fifth century). As in the west, these resulted in extensive destruction. But most of this area remained under Roman rule for several centuries longer than in the west. A key difference is that the eastern Roman empire was able to draw on the resources of a hinterland in Asia Minor, the Near East, and Egypt that was almost invulnerable to barbarians based in Europe. Constantinople, too, proved exceptionally resistant to siege. The eastern empire thus managed to sustain relative prosperity in Asia and Egypt and to tap their wealth to maintain imperial rule (albeit with less security and prosperity) in the Balkans and even to restore it temporarily to Italy and North Africa. At least for time: when, in the seventh century, most of the wealthy provinces of Asia and Africa were lost to Persian and then to Arab empires, the east Roman state lost some 4/5 of its revenues [63]. Most of Italy fell to the Lombards, and the Balkans to Avars and Slavs.

## Ecological complexities: Climate and disease

Another set of complications relates to the natural environment. Table 6, based on Harper [27], lists a number of natural disasters which may have affected the fate of Rome. Consider just one of these, the best documented and most thoroughly analyzed, the plague of Justinian. This is now securely identified as bubonic plague, *Yersinia pestis* [96, 97]. The first round of plague was apparently devastating, resulting in major population losses. The plague recurred for nearly two centuries before it disappeared. This period saw the failure of the eastern empire's initially successful campaigns to restore the empire in the west, and further massive territorial losses in the east. Elsewhere during this period, Anglo-Saxons probably pushed Britons to the margins of southern Britain. By the time the plague receded, the ethnic and linguistic map of Europe and the Mediterranean looked very different.

It is plausible that the plague contributed to these developments (but see [98]). However, it is important to note that the plague had the consequences that it did because it struck when it did. We don't have to turn to counterfactuals to support this assertion; nature ran the

**Table 6. Environmental shocks and the fate of Rome.**

| Years CE | Shock | Consequence |
|---|---|---|
| 165, 170s-190s | Antonine plague (smallpox?) | Loss of 10–20%? of empire's population, especially cities, army |
| 249–262, later? | Cyprian plague | Major population loss |
| 150 and following | End of Roman climatic optimum | Hard times |
| 300s | Drought on the steppes? | Huns move west? |
| 536, 540 | Volcanic eruptions | Coldest decade in 2000 years |
| 555–755 | Justinian plague (bubonic) | Massive population loss Europe-wide (25–50%?) |
| 530s-680s | Late Antique Little Ice Age (peak) | Hard times |

Harper [27]. See also recent findings on plague [98], smallpox [99], and climate [100].

experiment a second time in the late Middle Ages. When *Yersinia pestis* returned, starting with the 14th century Black Death, its direct effects—initial mass mortality, recurring epidemics over several centuries—were much the same. But in post-barbarian Europe, approaching Malthusian limits in some places after centuries of strong demographic and geographic expansion, the social and political sequelae of the plague were different. There was no large-scale redrawing of the ethno-linguistic map. There was no lasting collapse in social complexity, instead the socioeconomic aftermath in the West after a century was the waning of serfdom and the advance of urbanism.

## Overview

> There is a very real sense . . . in which the Roman Empire, in the long term, sowed the seeds of its own destruction. Its economic, military and diplomatic tentacles transformed adjacent populations until they were strong enough to rip it apart [18].

Europe began the first millennium CE out of equilibrium, with a steep gradient in resources and social complexity from core to periphery. Eventually a more even distribution was attained [18]. But the transition was conflict-ridden, driven by mismatches between wealth and power, with a sustained decline in resources and complexity as an unintended consequence.

In the Discussion section, we give some attention to how the *longue durée* [101] of the model relates to conjunctures over the medium term, how intermediate developments were mostly swept along by the long-term tide of barbarigenesis and its consequences, but sometimes swam against it. A reprise: Social complexity depends on the availability of resources. In first millennium Europe, an increase in resources, the result of internal development and diffusion, at times facilitated the growth of social complexity in both core and periphery. At other times, a decline in resources, the result in part of rent-seeking and collateral damage, led to a loss, or even collapse, of social complexity, most pronounced in the core, but also evident in intermediate areas. Core and periphery differed not only in the amount of resources they had but in how they allocated resources between production and fighting. The resulting wealth-power mismatch led initially to resource transfers and peripheral militarization within the Roman empire. It also led to raiding and plunder, to the expansion and consolidation of barbarian polities, and to population movements toward wealthier regions. Collateral damage resulted, both because of immediate destruction, and because successor polities geared toward rent seeking were mostly less effective at maintaining resources and social complexity over the long run.

There were complications and countertrends, of course. Geography was more irregular than in our model (but not decisively partitioned between steppe and sown as in other parts of Eurasia). And other phenomena are evident on a medium scale, including cycles of state making, and ecological shocks. Nevertheless (or so we argue) barbarigenesis among "people adjacent to a state but not in it" [16] and dealings between core and periphery stand out as a major engine of long run historical dynamics in first millennium Europe—and perhaps more generally.

## Conclusion

Our model may apply to other episodes of social collapse and recovery: the eastern Mediterranean from the twelfth century BCE [102], the classic Maya from the eighth century CE [103], Ancestral Puebloans from the thirteenth century CE [104], or mainland southeast Asia over

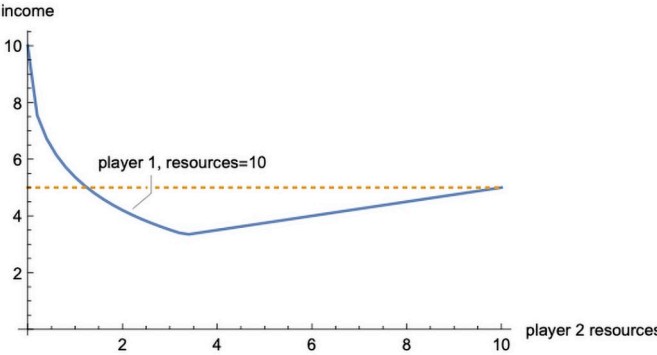

**8 a**

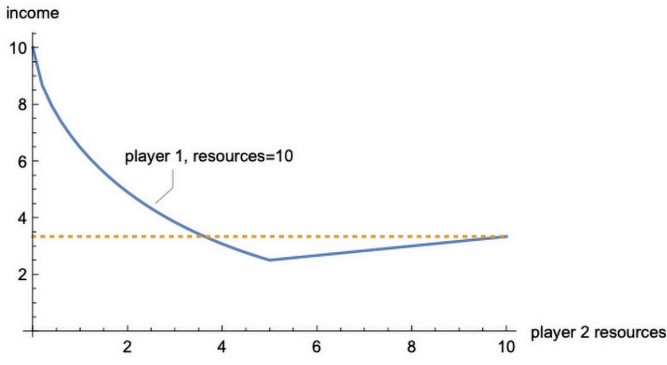

**8 b**

**Fig 8. Decisive battle and the end of the barbarian age.** a. (A reprise of Fig 1). Income in the two person game for player 1, resources = 10, facing player 2, varying resources, assuming less decisive "ancient" warfare, m = 1. Dashed line is income in an equal contest. b. Income in the two person game for player 1, resources = 10, facing player 2, varying resources, assuming more decisive "modern" warfare, m = 2. Dashed line is income in an equal contest.

the same period [105]. These are topics for another occasion. Here we consider instead a case of barbarigenesis and collapse that *didn't* happen.

In contemplating the decline and fall of the Roman empire, Gibbon [106] was led to "inquire with anxious curiosity, whether Europe is still threatened with a repetition of those calamities." We can cast his negative answer to this query in mathematical form.

Suppose we return to our first presentation of the paradox of power, Hirshleifer's two person game [19]. Eqs 3 and 4 include a parameter *m*, the decisiveness of conflict. Above, we let *m* = 1, so that the player who puts twice the effort into fighting gets twice the share of combined income. In this case the paradox of power operates strongly. This is seen in Fig 8a, which shows, for *m* = 1, the income accruing to player 1, resources = 10, facing an equal or weaker player, assuming optimal play. Over most of the range of resources, player 1 is worse off than if he faced an equal player. His income at its lowest is 1/3 lower than if player 2 were his equal.

Fig 8b shows the results for *m* = 2. Player 1's income when confronting an equal player is now less than it was for *m* = 1, because it pays for both sides to put more of their resources into fighting and less into production. But the paradox of power is also diminished: the relative

disadvantage of facing a weaker player is less, over a narrower range. The stronger player's income at its lowest is 1/4 lower than if player 2 were his equal. With more decisive conflict, there is less scope for barbarigenesis.

There are reasons to think that the military revolution of early modern Europe [107], especially the development of firearms, led to increasing military economies of scale in which "an increase of $x$% in all inputs increases an army's destructive capability by more than $x$%" [108]. In the present context, this translates into a higher value of $m$. An expected consequence, as shown in Fig 8, is the diminution or disappearance of the wealth-power mismatch between core and periphery which was a hallmark of the ancient world. Perhaps largely for this reason, the military relationship between the West and the Rest in modern times followed a very different course from the relationship between Roman and barbarian.

## Supporting information

**S1 File.**
(PDF)

**S2 File.**
(PDF)

**S3 File.**
(PDF)

**S4 File.**
(PDF)

## Acknowledgments

I thank Daniel Hoyer and two anonymous reviewers for advice that helped to improve this article.

## Author Contributions

**Conceptualization:** Doug Jones.

**Formal analysis:** Doug Jones.

**Investigation:** Doug Jones.

**Methodology:** Doug Jones.

**Software:** Doug Jones.

**Writing – original draft:** Doug Jones.

**Writing – review & editing:** Doug Jones.

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
