## [Decision Letter · Decision Letter 0]

12 Apr 2021

PONE-D-21-05786

Barbarigenesis and the collapse of complex societies: Rome and after

PLOS ONE

Dear Dr. Jones,

Thank you for submitting your manuscript to PLOS ONE. After careful consideration, we feel that it has merit but does not fully meet PLOS ONE’s publication criteria as it currently stands. Therefore, we invite you to submit a revised version of the manuscript that addresses the points raised during the review process.

All comments have to be addressed before re-submission.

We look forward to receiving your revised manuscript.

Kind regards,

Peter F. Biehl, PhD

Academic Editor

PLOS ONE

Journal Requirements:

PLOS requires an ORCID iD for the corresponding author in Editorial Manager on papers submitted after December 6th, 2016. Please ensure that you have an ORCID iD and that it is validated in Editorial Manager. To do this, go to ‘Update my Information’ (in the upper left-hand corner of the main menu), and click on the Fetch/Validate link next to the ORCID field. This will take you to the ORCID site and allow you to create a new iD or authenticate a pre-existing iD in Editorial Manager. Please see the following video for instructions on linking an ORCID iD to your Editorial Manager account: https://www.youtube.com/watch?v=_xcclfuvtxQ

We note that you have stated that you will provide repository information for your data at acceptance. Should your manuscript be accepted for publication, we will hold it until you provide the relevant accession numbers or DOIs necessary to access your data. If you wish to make changes to your Data Availability statement, please describe these changes in your cover letter and we will update your Data Availability statement to reflect the information you provide.

Please include captions for your Supporting Information files at the end of your manuscript, and update any in-text citations to match accordingly. Please see our Supporting Information guidelines for more information: http://journals.plos.org/plosone/s/supporting-information.

Additional Editor Comments:

Your manuscript has now been seen by two referees, whose comments are appended below. You will see from these comments that while the referees find your work of interest, they have raised substantial concerns that must be addressed. In light of these comments, we cannot accept the manuscript for publication, but would be interested in considering a revised version that addresses these concerns.

We hope you will find the referees' comments useful as you decide how to proceed. Should presentation of further data and analysis allow you to address these criticisms, we would be happy to look at a substantially revised manuscript. However, please bear in mind that we will be reluctant to approach the referees again in the absence of major revisions.

Reviewers' comments:

Reviewer's Responses to Questions

**Comments to the Author**

1. Is the manuscript technically sound, and do the data support the conclusions?

Reviewer #1: Yes

Reviewer #2: Partly

2. Has the statistical analysis been performed appropriately and rigorously? 

Reviewer #1: Yes

Reviewer #2: Yes

3. Have the authors made all data underlying the findings in their manuscript fully available?

Reviewer #1: Yes

Reviewer #2: Yes

4. Is the manuscript presented in an intelligible fashion and written in standard English?

Reviewer #1: Yes

Reviewer #2: Yes

5. Review Comments to the Author

Reviewer #1: Use of game-theory to elucidate 'barbarigenesis' (meaning 'decline in complexity') is innovative. Basic idea of mismatch between wealth and power as one of the driving forces behind barbarigenesis is insightful and plausible. Paper clearly deserves to be published. There is, however, some room for improvement. 1. Paper signals rise in taxation in Late Empire but does not explore dymanics lying behind this trend. As Peter Bang explains in Ch. 15 of The Oxford Handbook of the State in the Ancient Near East and Mediterranean (2013), the Roman empire started out as a CHEAP empire, in the sense that (initially) military expenditure was considerably lower than the total military expenditure of the empires conquered by the Romans. Between the 1st and 5th centuries CE this advantage was eroded; 2. The author's model works well for the western empire but less well for the East. In the East the Romans faced another tributary empire (initially the Parthians, subsequently the Sassanids) rather than a series of 'barbarian' societies. This produced a different dynamic in terms of mismatches between wealth and power. Arguably this difference helps to explain the greater longevity of the eastern half of the Roman empire (among other factors, of course); 3. The author links the partial collapse of the eastern half of the Roman empire to the Justinianic plague. While there is an element of truth in this, things were more complex. The loss of the Near East to the invading Arabs coincided with a severe weakening of the Roman (and Sassanid) empires as a result of a hugely expensive war between the two empires. Eastern-Roman reliance on a mobile army of perhaps 40,000 men to resolve major military crises created further vulnerabilities (Kaegi, Byzantium and the early Islamicc conquests, 2003). Since there was only ONE mobile army, destruction of this force created huge problems. It remains true that the decline in population and productive resources resulting from the Justininianic plague made (partial) recovery more diffiicult.

Reviewer #2: Overall Impressions:

Overall, this is an interesting piece. I appreciate the author’s use of game theory as a means to ‘measure’ competition among an empire and states or polities on its periphery, which has potential. At the same time, I cannot help but think that what is presented is a perhaps too simplistic (as opposed to simplified) model for state decline, in which a state (undefined in relation to the historical examples presented within the paper) can either choose to invest in ‘production’ (undefined) or ‘power’ (which should be presented as ‘military power’). Surely there are other factors at play here (in the historical context of the late Roman Empire, there is contagion, including the spread of malaria and various pandemics, decline in state bullion reserves, the collapse of the middle income class from the debasement of Rome’s coinage, the polarization of wealth, improvements in military technology and tactics among Rome’s neighbors and competitors/enemies (such as the Parthians/Sassanians and the Germans, to name but two), to name just a few). Some of this is acknowledged by the author, but dismissed as irrelevant in comparison to the interplay of production vs. power (this dismissal itself should be justified in greater detail). Also, I am very uncomfortable with the use of the term barbarigenesis, and the use of terms such as barbarian, civilization, sophisticated, etc. when applied to various historical groups (both figuratively with respect to the model and literally when the model is applied to the Roman Empire and its neighbors). I think that What is presented is an evolutionary model of human history, which seems many decades out of touch with contemporary historical, anthropological, and archaeological theory. The process of becoming a barbarian is one that I am unfamiliar with and, again, presents a historical dichotomy between ‘civilized’ and ‘barbaric’ or ‘uncivilized’ historical societies, states, groups, etc. The Greeks and Romans defined what the term meant, and this usage, to my knowledge, has not become less pejorative or ethnocentric over time. What an analysis like this masks, potentially, is an honest and relatively objective analysis of historical trends.

With respect to the mathematical model, this seems quite straightforward and is presented quite clearly but the author. When it comes to testing the model, they make their sources of data and the data itself clear, and the significance of that data, in particular the demographic data, to the overall historical question at hand. There are perhaps some other more recent studies on population and economic production that could be cited, but the scholarly consensus on general trends in population and productivity have not changed much over the past few decades. Still, more up-to-date bibliography would be welcome in a few instances.

Overall, however, the article comes to some very reasonable conclusions, when it turns to applying the model and to the discussion of the model in light of historical and archaeological evidence. The author acknowledges throughout that there are other factors not incorporated into the model, some of which may be causal, but only marginally.

Research question/design:

The research question, which proposes to use a specific method from game theory and apply this to competition among states/societies with a socially more complex, diverse, and geographically broad core (empire, perhaps) and peripheral groups with whom this central entity interacts in a variety of ways, is inherently interesting. As the author notes, the model they are using significantly simplifies complex historical realities, but that this does not disqualify the method as it appears to show results consistent with those historical realities and which may help explain them (in whole or in part). At the same time, and as is noted in my comments below, the model does not take into account other possible factors which may have contributed to the process (barbarigenesis, for which I suggest developing a different name) and been equally or more causal. There is no scope within the mathematical model as presented to include other factors (geographical, social reorganization, contagion, technological, etc.). Could the model be modified to include other factors, or to at least demonstrate causality for the two factors (production vs. power) presented in the paper?

Specific comments on ms:

Throughout, there is reference to the year 0. Is this intentional? Do I not understand the calendrical system being used herein? Should it not be the year 1?

Abstract: “This article develops a mathematical model of barbarigenesis – the formation of barbarian societies adjacent to more complex societies – and its consequences, and applies the model to the case of Europe in the first millennium CE.” What is meant here by ‘barbarian societies’? Presumably, these societies existed prior to the arrival of imperializing powers such as Rome, but were labelled as such by the imperializing powers themselves (returning to the Romans, in relation to their civilizing or ‘ordering’ mission as they sought to spread cosmos, good order in harmony with the gods, through the exercise of their imperium, military power). In this sense, there is no ‘genesis’ of barbarians in the sense presented in this article: they were already there. This term would make more sense if it were applied to the interaction and military conflict among an established imperial state such as Rome and its neighbors with less complex or hierarchical social organization, which led to military conflict (among other outcomes), and which, over time, changed the imperial center itself (perhaps through direct conflict of arms would fit in with the themes of this paper?). I think that what you want to describe is a real phenomenon related to collapse and transition. But, I think that the basic terms need to be better defined throughout to prevent confusion on the part of your reader (see below for more examples of this). Also, some terms should be reconsidered and replaced with more historically appropriate terms that are less ethnocentric and pejorative, that is biased toward the particular social group whose members have written the historical accounts of such periods of competition, conflict, decline, and transition (or partial collapse). 'civilized core': what makes the core 'civilized'? Is there a less charged term that could be used to describe the core in your model?

One of the terms that needs to be defined more clearly and whose usage needs to be better justified is Barbarigenesis, a term that I am not familiar with. Why use such a pejorative term? What constitutes a barbarian society? Why use this term and not one less loaded with meaning? Few contemporary historians or archaeologists working on Late Antiquity in Europe and the Mediterranean use this term in the manner it is used in the paper (or so I think, but, again, the term is not clearly defined, nor is the process whereby a peripheral society becomes 'barbaric'). As the central point of the paper is to provide a mathematical model to examine why this process exists and what drives it, a clear, detailed, yet concise definition of this term, and the term 'barbarian' would greatly improve the quality of the article. Other terms that could be defined more clearly and fully: 'wealth' 'power' (make it clear from the first usage that this term refers to the exercise of military or coercive state power). 'methods of economics': what methods in particular (line104)? 'production': this is an integral part of your formula, but it is not clearly defined, although a definition is hinted at throughout the paper; why not define it clearly at its first usage so that your reader knows exactly what you mean by this term with respect to both your model and to its historical application in the second part of your paper?

The paper mentions 'evolution' 'development' etc. a number of times throughout, putting it within the realm of a social evolutionary approach to human development (progress, as seen through the development of complex human societies, vs. regress, with a return to less complex social organization). Lines 70ff are a good example of this. I advocate using slightly different terms, or at least making this bias more overt to the reader.

The use of block quotes: Line 96, for example. I am not sure why so many of these appear throughout the text. They are not discussed in any detail and present material that does not appear to be well integrated into the overall text of the article.

Lines 292-296" Allocation of Resources. There seems to be a built-in assumption that at some point rent-seeking or unproductive diversion of resources into contests becomes prevalent. This point could be expanded upon within the historical example presented later on. Turning to that historical example, how did this manifest itself in the Late Roman Empire? As the author notes elsewhere, there was a significant economic recovery in the fifth century CE, but was the resulting system different with respect to the investment of resources in comparison to the first through third centuries CE? Did this reorganization and redistribution of wealth, in particular property, have an impact on production (you clearly seem to indicate that it does in both parts of your article), but perhaps a bit more detail would be welcome (how does this affect production; does this in and of itself potentially reduce the ability of a state to engage in conflict and, if so, how?).

Lines 386-387: These demographic sources are of significantly different value. As noted elsewhere, you might want to include some more recent publications on ancient demography, and to include work by Saskia Hin, Tim Parkin, and others. Lo Cascio has some more recent publications on Roman demography as well. Hin has a nice overview of the high vs. low count models and suggests a middle count as a solution, which really only problematizes things further.

Lines 389-391: Make more explicit in the text what is mean by the 'developed core of Western Eurasia.

Lines 482ff: You use demographic measures by country, which is how they are presented in your source material. This sounds really strange to historians and archaeologists of ancient and medieval Europe and the Mediterranean as these entities did not exist (aside from Italy, as an exception). Is there a more effective way to present this material so that it might have meaning to those who specialize in this part of the world and the aforementioned historical periods? Also, as is noted elsewhere, and as the author notes, McEvedy and Jones is a little dated and does not incorporate archaeological survey data, for example. While the general overall patterns appear to be consistent across sources, particularly across the Roman empire and central Europe, is this true for each region and sub-region?

Lines 505ff: the qualitative test of the model by reference to historical data/narratives. It is stated that there is broad support for the role of 'barbarians' in the decline and fall of the Roman empire, and the transition to Medieval western Europe (the recovery). Is this indeed true? Is the term 'barbarian' universally used among those working on this period today? Quoting Chris Wickham (lines 523 ff), it is noted that there is a dramatic economic simplification of most of the west, and the examples used to demonstrate this are building projects (presumably urban), artisanal production, and the increased in localized exchange. The reality is quite complex and highly dependent on where in the west the text is referencing and what specific time period we are talking about. Lines 516-517 note that this is true of the entire period 200 to 700, where we see a downward economic trend, but this is another quote. Is the author referring to the same time period in their analysis?

The section on opportunity costs, lines 622ff: This is a welcome inclusion.

The section on rent-seeking lines 680ff: This strikes me as an extremely important element in the institutional decline of the Roman system of governance and military. This is important for understanding both the decline of the Roman economy, but also the integration of non-Roman (typically Germanic) elements within the economic system in a way not seen before. In this sense; once this happens, the peripheral peoples beyond the frontier become integral to the empire within its frontiers. This is an important point. This point is addressed again in lines 840ff when the concept of frontiers or borders are discussed and it is noted that their inclusion might be a useful supplement to the model presented herein.

The section on ecological complexities: Some have argued that this is just as important, or perhaps even more important, that the economic and military issues discussed throughout this paper, so perhaps this should be acknowledged earlier on in the paper rather than tacking it on here at the end. To the list of environmental shocks can be added the increase in malaria.

6. PLOS authors have the option to publish the peer review history of their article (what does this mean?). If published, this will include your full peer review and any attached files.

Reviewer #1: No

Reviewer #2: No

---

## [Author Response · Author response to Decision Letter 0]

1 Jun 2021

I have also submitted the material below as an attached file, with formatting which is missing below.

To whom it may concern

Here is my response to reviews (reviews underlined, my responses not underlined, line numbers in brackets []). The comments were generally quite helpful, and I have made numerous changes in the article, including a pretty thorough rewrite of the introduction. In a few cases I disagree with comments (e.g. regarding the use of block quotes), and give my reasons.

Thank you for your consideration,

Doug Jones

Reviewer #1: Use of game-theory to elucidate 'barbarigenesis' (meaning 'decline in complexity') is innovative. Basic idea of mismatch between wealth and power as one of the driving forces behind barbarigenesis is insightful and plausible. Paper clearly deserves to be published. There is, however, some room for improvement. 1. Paper signals rise in taxation in Late Empire but does not explore dymanics lying behind this trend. As Peter Bang explains in Ch. 15 of The Oxford Handbook of the State in the Ancient Near East and Mediterranean (2013), the Roman empire started out as a CHEAP empire, in the sense that (initially) military expenditure was considerably lower than the total military expenditure of the empires conquered by the Romans. Between the 1st and 5th centuries CE this advantage was eroded; 2. The author's model works well for the western empire but less well for the East. In the East the Romans faced another tributary empire (initially the Parthians, subsequently the Sassanids) rather than a series of 'barbarian' societies. This produced a different dynamic in terms of mismatches between wealth and power. Arguably this difference helps to explain the greater longevity of the eastern half of the Roman empire (among other factors, of course); 3. The author links the partial collapse of the eastern half of the Roman empire to the Justinianic plague. While there is an element of truth in this, things were more complex. The loss of the Near East to the invading Arabs coincided with a severe weakening of the Roman (and Sassanid) empires as a result of a hugely expensive war between the two empires. Eastern-Roman reliance on a mobile army of perhaps 40,000 men to resolve major military crises created further vulnerabilities (Kaegi, Byzantium and the early Islamicc conquests, 2003). Since there was only ONE mobile army, destruction of this force created huge problems. It remains true that the decline in population and productive resources resulting from the Justininianic plague made (partial) recovery more difficult.

1. I have added a paragraph discussing Rome’s advantages over Hellenistic empires [850] 2. I discuss the eastern Roman empire in [1057]. I agree that the story outside Europe – contests between rival empires – falls outside the model. But the eastern empire in Europe did face, and ultimately fall to, barbarian (Sclavenian, Avar, Bulgar) invaders. Having a protected hinterland in Asia Minor delayed but didn’t prevent this. 3. Agreed, more was going on in the 6th -7th C empire than just the Justiniac plague. However, many authors have emphasized the role of the plague in the partial collapse of the Eastern empire, so it is appropriate that I bring this up. (I also cite recent evidence that the plague was less disruptive than sometimes suggested.) 

Reviewer #2: Overall Impressions:

Overall, this is an interesting piece. I appreciate the author’s use of game theory as a means to ‘measure’ competition among an empire and states or polities on its periphery, which has potential. At the same time, I cannot help but think that what is presented is a perhaps too simplistic (as opposed to simplified) model for state decline, in which a state (undefined in relation to the historical examples presented within the paper) can either choose to invest in ‘production’ (undefined) or ‘power’ (which should be presented as ‘military power’). Surely there are other factors at play here (in the historical context of the late Roman Empire, there is contagion, including the spread of malaria and various pandemics, decline in state bullion reserves, the collapse of the middle income class from the debasement of Rome’s coinage, the polarization of wealth, improvements in military technology and tactics among Rome’s neighbors and competitors/enemies (such as the Parthians/Sassanians and the Germans, to name but two), to name just a few). Some of this is acknowledged by the author, but dismissed as irrelevant in comparison to the interplay of production vs. power (this dismissal itself should be justified in greater detail). Also, I am very uncomfortable with the use of the term barbarigenesis, and the use of terms such as barbarian, civilization, sophisticated, etc. when applied to various historical groups (both figuratively with respect to the model and literally when the model is applied to the Roman Empire and its neighbors). I think that What is presented is an evolutionary model of human history, which seems many decades out of touch with contemporary historical, anthropological, and archaeological theory. The process of becoming a barbarian is one that I am unfamiliar with and, again, presents a historical dichotomy between ‘civilized’ and ‘barbaric’ or ‘uncivilized’ historical societies, states, groups, etc. The Greeks and Romans defined what the term meant, and this usage, to my knowledge, has not become less pejorative or ethnocentric over time. What an analysis like this masks, potentially, is an honest and relatively objective analysis of historical trends.

I’ve revised the introduction extensively [37-164], including more discussion of terms (production, power). This includes citations of the literature in economics and economic history, connecting the present discussion with theories of rent-seeking and protection rent [89ff].

Yes, there are other factors at play, especially in the late Roman empire. I’ve added some discussion here [759ff], a paragraph on what is admittedly a huge topic. I think the following is justified: (1) a growing threat of barbarian invasion was arguably (not certainly) a factor in the 3rd C crisis, and in some of the changes between Principate and Dominate. But there were other things going on. (2) Internal crises alone were not going to bring down the Western empire in the 5th C, or the Eastern empire in Europe in the 7th. Barbarian invasions were crucial (Hence all the block quotes meant to show that this is a pretty standard view.)

Regarding the terms “barbarian,” “civilized,” etc. more below.

On one specific topic, Reviewer 2 writes, I think that What is presented is an evolutionary model of human history, which seems many decades out of touch with contemporary historical, anthropological, and archaeological theory. It is true that significant numbers of historians and anthropologists lean toward historical particularism, and shy away from talking about social complexity and social scale and trends in social evolution over time, Here’s a quote from Joyce Marcus in her review article on social evolution in archeology, making the case for thinking about social evolution: “Archaeologists … need to maintain an ongoing dialogue with ethnologists and ethnohistorians …. This collaboration has become increasingly difficult owing to many current anthropologists’ antipathy toward generalization, controlled comparison, and the search for universal patterns.” But there are also plenty of people nowadays working on these topics, and the introduction cites a number of them. Jared Diamond has reached a broad audience with popular books on the evolution of social complexity, and collapse. These topics also deserve scholarly treatment, and they are getting it from the authors cited, among others.

With respect to the mathematical model, this seems quite straightforward and is presented quite clearly but the author. When it comes to testing the model, they make their sources of data and the data itself clear, and the significance of that data, in particular the demographic data, to the overall historical question at hand. There are perhaps some other more recent studies on population and economic production that could be cited, but the scholarly consensus on general trends in population and productivity have not changed much over the past few decades. Still, more up-to-date bibliography would be welcome in a few instances.

I’ve added some references as suggested, see below.

Overall, however, the article comes to some very reasonable conclusions, when it turns to applying the model and to the discussion of the model in light of historical and archaeological evidence. The author acknowledges throughout that there are other factors not incorporated into the model, some of which may be causal, but only marginally.

I wouldn’t say that these other factors are marginal. Some of them are important in the short to medium term. But I suggest that they tend to average out over the very long run.

Research question/design:

The research question, which proposes to use a specific method from game theory and apply this to competition among states/societies with a socially more complex, diverse, and geographically broad core (empire, perhaps) and peripheral groups with whom this central entity interacts in a variety of ways, is inherently interesting. As the author notes, the model they are using significantly simplifies complex historical realities, but that this does not disqualify the method as it appears to show results consistent with those historical realities and which may help explain them (in whole or in part). At the same time, and as is noted in my comments below, the model does not take into account other possible factors which may have contributed to the process (barbarigenesis, for which I suggest developing a different name) and been equally or more causal. There is no scope within the mathematical model as presented to include other factors (geographical, social reorganization, contagion, technological, etc.). Could the model be modified to include other factors, or to at least demonstrate causality for the two factors (production vs. power) presented in the paper?

As I discuss in the Introduction [156-164]and Discussion [10010-1042, 1111-1130], there are models of, e.g., regular “secular cycles” and “imperiogenesis” that maybe capture some political ups-and-downs on a scale of a few centuries. What these don’t do is account for the large scale collapse and recovery that is characteristic of 1st millennium Europe. It might be possible to devise more complicated models (e.g. agent based) that would deal with trends and cycles at both scales. Also, incorporating geographic variation, especially the distinction between Steppe and Sown is an obvious next step for extending the model to other parts of Eurasia [1048-1056]. All this has to be something for the future.

Specific comments on ms:

Throughout, there is reference to the year 0. Is this intentional? Do I not understand the calendrical system being used herein? Should it not be the year 1?

I have changed this to the year 1 throughout the text. I have left it as 0 in some of the figures, where it is more computationally convenient. I have added a note in the figure captions to that effect.

Abstract: “This article develops a mathematical model of barbarigenesis – the formation of barbarian societies adjacent to more complex societies – and its consequences, and applies the model to the case of Europe in the first millennium CE.” What is meant here by ‘barbarian societies’? Presumably, these societies existed prior to the arrival of imperializing powers such as Rome, but were labelled as such by the imperializing powers themselves (returning to the Romans, in relation to their civilizing or ‘ordering’ mission as they sought to spread cosmos, good order in harmony with the gods, through the exercise of their imperium, military power). In this sense, there is no ‘genesis’ of barbarians in the sense presented in this article: they were already there. This term would make more sense if it were applied to the interaction and military conflict among an established imperial state such as Rome and its neighbors with less complex or hierarchical social organization, which led to military conflict (among other outcomes), and which, over time, changed the imperial center itself (perhaps through direct conflict of arms would fit in with the themes of this paper?). I think that what you want to describe is a real phenomenon related to collapse and transition. But, I think that the basic terms need to be better defined throughout to prevent confusion on the part of your reader (see below for more examples of this). Also, some terms should be reconsidered and replaced with more historically appropriate terms that are less ethnocentric and pejorative, that is biased toward the particular social group whose members have written the historical accounts of such periods of competition, conflict, decline, and transition (or partial collapse). 'civilized core': what makes the core 'civilized'? Is there a less charged term that could be used to describe the core in your model?

One of the terms that needs to be defined more clearly and whose usage needs to be better justified is Barbarigenesis, a term that I am not familiar with. Why use such a pejorative term? What constitutes a barbarian society? Why use this term and not one less loaded with meaning? Few contemporary historians or archaeologists working on Late Antiquity in Europe and the Mediterranean use this term in the manner it is used in the paper (or so I think, but, again, the term is not clearly defined, nor is the process whereby a peripheral society becomes 'barbaric'). As the central point of the paper is to provide a mathematical model to examine why this process exists and what drives it, a clear, detailed, yet concise definition of this term, and the term 'barbarian' would greatly improve the quality of the article. Other terms that could be defined more clearly and fully: 'wealth' 'power' (make it clear from the first usage that this term refers to the exercise of military or coercive state power). 'methods of economics': what methods in particular (line104)? 'production': this is an integral part of your formula, but it is not clearly defined, although a definition is hinted at throughout the paper; why not define it clearly at its first usage so that your reader knows exactly what you mean by this term with respect to both your model and to its historical application in the second part of your paper?

Regarding the terms “barbarian,” “barbarigenesis.” Among historians of late Antiquity, some avoid the term barbarian. Others use it freely (e.g. Harper in The Fate of Rome). A number of historians, including Heather, Halsall, Mathisen, Goffart, and Thompson (see references) use the term in book titles. So my using the expression is not outside the bounds of current scholarly convention. I’ve tried to think of alternatives, but have had trouble coming up with much. “Tribal” or “non-state” doesn’t really cover it: some of these populations had (or took over) kingdoms for some of the time. “Germanic” again leaves out Slavs and Huns and such. “Peripheral” groups works sometimes, but less well when they move from periphery to core; also really remote peripheral groups fall outside the definition of barbarism used here. A measure of the difficulty is apparent in Wickham’s writing: he uses ”barbarian” a lot, while distancing himself from the term by putting it in scare quotes, which a lot of style manuals would recommend against. The best I’ve been able to come up with is (1) putting “barbarian” in quotes up to the point I define it, and thereafter losing the quotes, and (2) explicitly setting out reasons for using the term, and connecting it with the model of wealth-power mismatch. Note that this use is close to Scott’s. 

“Barbarigenesis” is a neologism, but it makes sense and I hope is useful given my (and Scott’s, and Owen Lattimore’s and di Cosmo’s) ideas about how some folk get to be barbarians. Reviewer 2: This term would make more sense if it were applied to the interaction and military conflict among an established imperial state such as Rome and its neighbors with less complex or hierarchical social organization, which led to military conflict (among other outcomes), and which, over time, changed the imperial center itself (perhaps through direct conflict of arms would fit in with the themes of this paper?). I absolutely agree, and have spelled this out.

Re: “civilized core,” I’ve mostly replaced this with just “core.” “Civilization” now appears only in quotations from other authors, and “sophisticated” appears only in a quotation from Scheidel.

I’ve spelled out more what I mean by “production,” “wealth” and “power” in xx, aligning it with work in economics and economic history on rent-seeking and protection rents [89-97].

The paper mentions 'evolution' 'development' etc. a number of times throughout, putting it within the realm of a social evolutionary approach to human development (progress, as seen through the development of complex human societies, vs. regress, with a return to less complex social organization). Lines 70ff are a good example of this. I advocate using slightly different terms, or at least making this bias more overt to the reader.

I lead off at the very beginning of the Introduction [37-46] by introducing the idea of social development, and citing a number of authors who make use of the this and related concepts. I make clear right away (second sentence) that social development may not mean a better life for the average person.

The use of block quotes: Line 96, for example. I am not sure why so many of these appear throughout the text. They are not discussed in any detail and present material that does not appear to be well integrated into the overall text of the article.

Here I disagree. Given the state of the field¬ – lots of controversies and disagreement – it seems important to document that views presented here are not just my own idiosyncratic notions, but have support from other scholars. For example, Reviewer 2 questions whether there is broad support for the role of 'barbarians' in the decline and fall of the Roman empire. The block quotes in [680-699] are meant to establish that multiple scholars with otherwise differing views agree on this point.

Lines 292-296" Allocation of Resources. There seems to be a built-in assumption that at some point rent-seeking or unproductive diversion of resources into contests becomes prevalent. This point could be expanded upon within the historical example presented later on. Turning to that historical example, how did this manifest itself in the Late Roman Empire? As the author notes elsewhere, there was a significant economic recovery in the fifth century CE, but was the resulting system different with respect to the investment of resources in comparison to the first through third centuries CE? Did this reorganization and redistribution of wealth, in particular property, have an impact on production (you clearly seem to indicate that it does in both parts of your article), but perhaps a bit more detail would be welcome (how does this affect production; does this in and of itself potentially reduce the ability of a state to engage in conflict and, if so, how?).

As set out in [731-741], it looks like there was an increase in the size of the military and in taxation in the 5th C relative to centuries 1-3, which is plausibly in part a response to increased barbarian pressure. The earlier view (from AHM Jones) that the empire was being suffocated by overtaxation doesn’t seem to be supported by archeology, but there does seem to have been an economic hit, and it does seem that elites in the later period were withdrawing some of their support for the empire in the face of the heavy burdens it imposed. Much more could be said, but I have made these basic points in the exposition.

Lines 386-387: These demographic sources are of significantly different value. As noted elsewhere, you might want to include some more recent publications on ancient demography, and to include work by Saskia Hin, Tim Parkin, and others. Lo Cascio has some more recent publications on Roman demography as well. Hin has a nice overview of the high vs. low count models and suggests a middle count as a solution, which really only problematizes things further.

I have consulted Hin and added reference to her work. I haven’t been able to get ahold of Parkin; my library has been dilatory in getting it for me.

Lines 389-391: Make more explicit in the text what is mean by the 'developed core of Western Eurasia.

This is Morris’s term; I have added material to spell out how he uses it.

Lines 482ff: You use demographic measures by country, which is how they are presented in your source material. This sounds really strange to historians and archaeologists of ancient and medieval Europe and the Mediterranean as these entities did not exist (aside from Italy, as an exception). Is there a more effective way to present this material so that it might have meaning to those who specialize in this part of the world and the aforementioned historical periods? Also, as is noted elsewhere, and as the author notes, McEvedy and Jones is a little dated and does not incorporate archaeological survey data, for example. While the general overall patterns appear to be consistent across sources, particularly across the Roman empire and central Europe, is this true for each region and sub-region?

Other scholars working on long-term economic development face the same issue of countries changing borders over time, and have adopted the same convention of just plowing ahead with modern countries. For example

https://www.nber.org/system/files/working_papers/w8460/w8460.pdf

I haven’t attempted any wholesale revision of McEvedy and Jones. They certainly need to be updated. As noted they seem to be consistent with other estimates. They may err on the low side, but the ratios of different populations over time and between countries are fairly similar, which is what matters for the data analysis. In a few cases that I’m familiar with (notably England) they may be off by a fairly large amount, and there is some discussion of this in the paper.

Lines 505ff: the qualitative test of the model by reference to historical data/narratives. It is stated that there is broad support for the role of 'barbarians' in the decline and fall of the Roman empire, and the transition to Medieval western Europe (the recovery). Is this indeed true? Is the term 'barbarian' universally used among those working on this period today? Quoting Chris Wickham (lines 523 ff), it is noted that there is a dramatic economic simplification of most of the west, and the examples used to demonstrate this are building projects (presumably urban), artisanal production, and the increased in localized exchange. The reality is quite complex and highly dependent on where in the west the text is referencing and what specific time period we are talking about. Lines 516-517 note that this is true of the entire period 200 to 700, where we see a downward economic trend, but this is another quote. Is the author referring to the same time period in their analysis?

I believe that there is broad support for the role of 'barbarians' in the decline and fall of the Roman empire, and that (with all due allowance for complications and countertrends in some places) the archeology shows clearly that there is a dramatic economic simplification of most of the west. I have included a number of block quotes [657-670, 681-699] to establish that this is not just my idiosyncratic reading of the evidence.

The section on opportunity costs, lines 622ff: This is a welcome inclusion.

The section on rent-seeking lines 680ff: This strikes me as an extremely important element in the institutional decline of the Roman system of governance and military. This is important for understanding both the decline of the Roman economy, but also the integration of non-Roman (typically Germanic) elements within the economic system in a way not seen before. In this sense; once this happens, the peripheral peoples beyond the frontier become integral to the empire within its frontiers. This is an important point. This point is addressed again in lines 840ff when the concept of frontiers or borders are discussed and it is noted that their inclusion might be a useful supplement to the model presented herein.

The section on ecological complexities: Some have argued that this is just as important, or perhaps even more important, that the economic and military issues discussed throughout this paper, so perhaps this should be acknowledged earlier on in the paper rather than tacking it on here at the end. To the list of environmental shocks can be added the increase in malaria.

Ecological complexities: there is now an acknowledgement of this point early on [156ff]

Malaria: My reading of the major work on the topic (Sallares, Malaria and Rome, 2002) is that malaria was a problem from early in Roman history, exacerbated by forest clearance, and a continuing problem right up to the twentieth century, not specifically a major contribution to decline.

6. PLOS authors have the option to publish the peer review history of their article (what does this mean?). If published, this will include your full peer review and any attached files.

Do you want your identity to be public for this peer review? For information about this choice, including consent withdrawal, please see our Privacy Policy.

Reviewer #1: No

Reviewer #2: No

---

## [Decision Letter · Decision Letter 1]

23 Jun 2021

Barbarigenesis and the collapse of complex societies: Rome and after

PONE-D-21-05786R1

Dear Dr. Jones,

We’re pleased to inform you that your manuscript has been judged scientifically suitable for publication and will be formally accepted for publication once it meets all outstanding technical requirements.

Kind regards,

Peter F. Biehl, PhD

Academic Editor

PLOS ONE

Additional Editor Comments (optional):

Reviewers' comments:

Reviewer's Responses to Questions

**Comments to the Author**

1. If the authors have adequately addressed your comments raised in a previous round of review and you feel that this manuscript is now acceptable for publication, you may indicate that here to bypass the “Comments to the Author” section, enter your conflict of interest statement in the “Confidential to Editor” section, and submit your "Accept" recommendation.

Reviewer #2: All comments have been addressed

2. Is the manuscript technically sound, and do the data support the conclusions?

Reviewer #2: Yes

3. Has the statistical analysis been performed appropriately and rigorously? 

Reviewer #2: Yes

4. Have the authors made all data underlying the findings in their manuscript fully available?

Reviewer #2: Yes

5. Is the manuscript presented in an intelligible fashion and written in standard English?

Reviewer #2: Yes

6. Review Comments to the Author

Reviewer #2: I think that the author has done an excellent job revising the article. All of my comments and concerns have been addressed clearly and decisively. As a result, I have very little to say at this point.

I encourage the author in future publications to present the various factors contributing to social/political/military change between core and periphery (opportunity costs, geography, environmental factors, etc.) in a single graphic. I had not thought of this in my original review, so it is unfair to note this in relation to this submission, but I recently reviewed an article on a similar topic (resilience in the western Empire during the Late Antique period) which used such a graphic (a large graph by century with each factor included on the timeline, including historical events, rainfall data, temperature, demographic change, etc.). It would be an effective way to represent some of the elements of the model discussed herein (in the Discussion section). This might expand on Table 6. It would not replace the text, but be a nice companion to it. Again, not a criticism of the submission, just advice for the author if they wish to continue to publish on this topic/model. And, if the editors of PLOS ONE would allow time for it, such a table would make a great addition to this submission.

The model itself is quite clear, and I think that this is the key factor for the entire submission. Even the most algorithmically-averse historian or archaeologist should be able to figure it out without much difficulty.

7. PLOS authors have the option to publish the peer review history of their article (what does this mean?). If published, this will include your full peer review and any attached files.

Reviewer #2: **Yes: **Myles C McCallum

---

## [Editor Report · Acceptance letter]

3 Sep 2021

PONE-D-21-05786R1 

Barbarigenesis and the collapse of complex societies: Rome and after 

Dear Dr. Jones:

I'm pleased to inform you that your manuscript has been deemed suitable for publication in PLOS ONE. Congratulations! Your manuscript is now with our production department. 

Kind regards, 

on behalf of

Dr. Peter F. Biehl 

Academic Editor

PLOS ONE